# Towards Robust Offline Reinforcement Learning under Diverse Data Corruption

**Rui Yang**[1]***, Han Zhong**[2]*, **Jiawei Xu**[3]*, **Amy Zhang**[4], **Chongjie Zhang**[5], **Lei Han**[6], **Tong Zhang**[1]
[1]HKUST,  [2]Peking University, [3]CUHK (SZ),  [4]University of Texas at Austin
[5]Washington University (St. Louis),  [6]Tencent Robotics X

## Abstract

Offline reinforcement learning (RL) presents a promising approach for learning reinforced policies from offline datasets without the need for costly or unsafe interactions with the environment. However, datasets collected by humans in real-world environments are often noisy and may even be maliciously corrupted, which can significantly degrade the performance of offline RL. In this work, we first investigate the performance of current offline RL algorithms under comprehensive data corruption, including states, actions, rewards, and dynamics. Our extensive experiments reveal that implicit Q-learning (IQL) demonstrates remarkable resilience to data corruption among various offline RL algorithms. Furthermore, we conduct both empirical and theoretical analyses to understand IQL's robust performance, identifying its supervised policy learning scheme as the key factor. Despite its relative robustness, IQL still suffers from heavy-tail targets of Q functions under dynamics corruption. To tackle this challenge, we draw inspiration from robust statistics to employ the Huber loss to handle the heavy-tailedness and utilize quantile estimators to balance penalization for corrupted data and learning stability. By incorporating these simple yet effective modifications into IQL, we propose a more robust offline RL approach named Robust IQL (RIQL). Extensive experiments demonstrate that RIQL exhibits highly robust performance when subjected to diverse data corruption scenarios.

## 1 Introduction

Offline reinforcement learning (RL) is an increasingly prominent paradigm for learning decision-making from offline datasets (Levine et al., 2020; Fujimoto et al., 2019; Kumar et al., 2019; 2020). Its primary objective is to address the limitations of RL, which traditionally necessitates a large amount of online interactions with the environment. Despite its advantages, offline RL suffers from the challenges of distribution shift between the learned policy and the data distribution (Levine et al., 2020). To address such issue, offline RL algorithms either enforce policy constraint (Wang et al., 2018; Fujimoto et al., 2019; Fujimoto & Gu, 2021; Kostrikov et al., 2021) or penalize values of out-of-distribution (OOD) actions (Kumar et al., 2020; An et al., 2021; Bai et al., 2022; Ghasemipour et al., 2022) to ensure the learned policy remains closely aligned with the training distribution.

Most previous studies on offline RL have primarily focused on simulation tasks, where data collection is relatively accurate. However, in real-world environments, data collected by humans or sensors can be subject to random noise or even malicious corruption. For example, during RLHF data collection, annotators may inadvertently or deliberately provide incorrect responses or assign higher rewards for harmful responses. This introduces challenges when applying offline RL algorithms, as constraining the policy to the corrupted data distribution can lead to a significant decrease in performance or a deviation from the original objectives. This limitation can restrict the applicability of offline RL to many real-world scenarios (Wang et al., 2024; Sun, 2023). To the best of our knowledge, the problem of robustly learning an RL policy from corrupted offline datasets remains relatively unexplored. It is important to note that this setting differs from many prior works on robust offline RL (Shi & Chi, 2022; Yang et al., 2022a; Blanchet et al., 2023), which mainly

---

*Equal Contribution. Correspondence to Rui Yang ⟨yangrui.thu2015@gmail.com⟩ and Han Zhong ⟨hanzhong@stu.pku.edu.cn⟩. Our code is available at https://github.com/YangRui2015/RIQL.

Figure 1: Performance of offline RL algorithms under random attacks on the Hopper task. Many offline RL algorithms are susceptible to different types of data corruption. Notably, IQL demonstrates superior resilience to 3 out of 4 types of data corruption.

focus on testing-time robustness, i.e., learning from clean data and defending against attacks during evaluation. The most closely related works are Zhang et al. (2022); Ye et al. (2023b); Wu et al. (2022), which focus on the statistical robustness and stability certification of offline RL under data corruption, respectively. Different from these studies, we aim to develop an offline RL algorithm that is robust to diverse data corruption scenarios.

In this paper, we initially investigate the performance of various offline RL algorithms when confronted with data corruption of all four elements in the dataset: states, actions, rewards, and dynamics. As illustrated in Figure 1, our observations reveal two findings: (1) current SOTA pessimism-based offline RL algorithms, such as EDAC (An et al., 2021) and MSG (Ghasemipour et al., 2022), experience a significant performance drop when subjected to data corruption. (2) Conversely, IQL (Kostrikov et al., 2021) demonstrates greater robustness against diverse data corruption. Further analysis indicates that the supervised learning scheme is the key to its resilience. This intriguing finding highlights the superiority of weighted imitation learning (Wang et al., 2018; Peng et al., 2019; Kostrikov et al., 2021) for offline RL under data corruption. From a theoretical perspective, we prove that the corrupted dataset impacts IQL by introducing two types of errors: (i) intrinsic imitation errors, which are the generalization errors of supervised learning objectives and typically diminish as the dataset size $N$ increases; and (ii) corruption errors at a rate of $\mathcal{O}(\sqrt{\zeta/N})$, where $\zeta$ is the cumulative corruption level (see Assumption 1). This result underscores that, under the mild assumption that $\zeta = o(N)$, IQL exhibits robustness in the face of diverse data corruption scenarios.

Although IQL outperforms other algorithms under many types of data corruption, it still suffers significant performance degradation, especially when faced with dynamics corruption. To enhance its overall robustness, we propose three improvements: observation normalization, the Huber loss, and the quantile Q estimators. We note that the heavy-tailedness in the value targets under dynamics corruption greatly impacts IQL's performance. To mitigate this issue, we draw inspiration from robust statistics and employ the Huber loss (Huber, 1992; 2004) to robustly learn value functions. Moreover, we find the Clipped Double Q-learning (CDQ) trick (Fujimoto et al., 2018), while effective in penalizing corrupted data, lacks stability under dynamics corruption. To balance penalizing corrupted data and enhancing learning stability, we introduce quantile Q estimators with an ensemble of Q functions. By integrating all of these modifications into IQL, we present Robust IQL (RIQL), a simple yet remarkably robust offline RL algorithm. Through extensive experiments, we demonstrate that RIQL consistently exhibits robust performance across various data corruption scenarios. We believe our study not only provides valuable insights for future robust offline RL research but also lays the groundwork for addressing data corruption in more realistic situations.

## 2 RELATED WORKS

**Offline RL.** To address the distribution shift problem, offline RL algorithms typically fall into policy constraints-based (Wang et al., 2018; Peng et al., 2019; Fujimoto & Gu, 2021; Kostrikov et al., 2021; Sun et al., 2024; Xu et al., 2023), and pessimism-based approaches (Kumar et al., 2020; An et al., 2021; Bai et al., 2022; Ghasemipour et al., 2022; Nikulin et al., 2023). Among these algorithms, ensemble-based approaches (An et al., 2021; Ghasemipour et al., 2022) demonstrate superior performance by estimating the lower-confidence bound (LCB) of Q values for OOD actions. Additionally, weighted imitation-based algorithms (Kostrikov et al., 2021; Xu et al., 2023) enjoy better simplicity and stability compared to pessimism-based approaches.

**Robust Offline RL.** In the robust offline setting, a number of works focus on testing-time or the distributional robustness (Zhou et al., 2021; Shi & Chi, 2022; Yang et al., 2022a; Panaganti et al.,

2022; Blanchet et al., 2023). Regarding the training-time robustness of offline RL, Li et al. (2023) investigate various reward attacks in offline RL. From a theoretical perspective, Zhang et al. (2022) study offline RL under data contamination. Different from these works, we propose an algorithm that is both provable and practical under diverse data corruption on all elements. More comprehensive related works are deferred to Appendix A.

## 3 PRELIMINARIES

**Reinforcement Learning (RL).** RL is generally represented as a Markov Decision Process (MDP) defined by a tuple $(S, A, P, r, \gamma)$. The tuple consists of a state space $S$, an action space $A$, a transition function $P$, a reward function $r$, and a discount factor $\gamma \in [0, 1)$. For simplicity, we assume that the reward function is deterministic and bounded $|r(s, a)| \leq R_{\max}$ for any $(s, a) \in S \times A$. The objective of an RL agent is to learn a policy $\pi(a|s)$ that maximizes the expected cumulative return: $\max_\pi \mathbb{E}_{s_0 \sim \rho_0, a_t \sim \pi(\cdot|s_t), s_{t+1} \sim P(\cdot|s_t, a_t)} [\sum_{t=0}^\infty \gamma^t r(s_t, a_t)]$, where $\rho_0$ is the distribution of initial states. The value functions are defined as $V^\pi(s) = \mathbb{E}_{\pi, P} [\sum_{t=0}^\infty \gamma^t r(s_t, a_t)|s_0 = s]$ and $Q^\pi(s, a) = r(s, a) + \gamma \mathbb{E}_{s' \sim P(\cdot|s, a)} [V^\pi(s')]$.

**Offline RL and IQL.** We focus on offline RL, which aims to learn a near-optimal policy from a static dataset $\mathcal{D}$. IQL (Kostrikov et al., 2021) employs expectile regression to learn the value functions:

$$\mathcal{L}_Q(\theta) = \mathbb{E}_{(s,a,s') \sim \mathcal{D}} \left[ (r(s, a) + \gamma V_\psi(s') - Q_\theta(s, a))^2 \right], \tag{1}$$

$$\mathcal{L}_V(\psi) = \mathbb{E}_{(s,a) \sim \mathcal{D}} \left[ \mathcal{L}_2^\tau(Q_\theta(s, a) - V_\psi(s)) \right], \quad \mathcal{L}_2^\tau(x) = |\tau - \mathbf{1}(x < 0)|x^2. \tag{2}$$

IQL further extracts the policy using weighted imitation learning with a hyperparameter $\beta$:

$$\mathcal{L}_\pi(\phi) = \mathbb{E}_{(s,a) \sim \mathcal{D}}[\exp(\beta \cdot A(s, a)) \log \pi_\phi(a|s)], \quad A(s, a) = Q_\theta(s, a) - V_\psi(s). \tag{3}$$

**Clean Data and Corrupted Data.** We assume that the uncorrupted data $(s, a, r, s')$ follows the distribution $(s, a) \sim \mu(\cdot, \cdot)$, $r = r(s, a)$, and $s' \sim P(\cdot \mid s, a)$. Here $\mu(\cdot, \cdot)$ is the behavior distribution. Besides, we use $\pi_\mu(a \mid s)$ to denote the conditional distribution. For the corrupted dataset $\mathcal{D}$, we assume that $\mathcal{D} = \{(s_i, a_i, r_i, s'_i)\}_{i=1}^N$ consists of $N$ samples, where we allow $(s_i, a_i)$ not to be sampled from the behavior distribution $\mu(\cdot, \cdot)$, $r_i = \tilde{r}(s_i, a_i)$, and $s'_i \sim \tilde{P}(\cdot \mid s_i, a_i)$. Here $\tilde{r}$ and $\tilde{P}$ are the corrupted reward function and transition dynamics for the $i$-th data, respectively. For ease of presentation, we denote the empirical state distribution and empirical state-action distribution as $d_\mathcal{D}(s)$ and $d_\mathcal{D}(s, a)$ respectively, while the conditional distribution is represented by $\pi_\mathcal{D}(a|s)$, i.e., $\pi_\mathcal{D}(a|s) = \frac{\sum_{i=1}^N \mathbf{1}\{s_i = s, a_i = a\}}{\max\{1, \sum_{i=1}^N \mathbf{1}\{s_i = s\}\}}$. Also, we introduce the following notations:

$$[\mathcal{T}V](s, a) = r(s, a) + \mathbb{E}_{s' \sim P(\cdot|s, a)}[V(s')], \quad [\tilde{\mathcal{T}}V](s, a) = \tilde{r}(s, a) + \mathbb{E}_{s' \sim \tilde{P}(\cdot|s, a)}[V(s')], \tag{4}$$

for any $(s, a) \in S \times A$ and $V : S \mapsto [0, R_{\max}/(1 - \gamma)]$.

## 4 OFFLINE RL UNDER DIVERSE DATA CORRUPTION

In this section, we first compare various offline RL algorithms in the context of data corruption. Subsequently, drawing on empirical observations, we delve into a theoretical understanding of the provable robustness of weighted imitation learning methods under diverse data corruption.

### 4.1 EMPIRICAL OBSERVATION

Initially, we develop random attacks on four elements: states, actions, rewards, and next-states (or dynamics). We sample $30\%$ of the transitions from the dataset and modify them by incorporating random noise at a scale of 1 standard deviation. Details about data corruption are provided in Appendix D.1. As illustrated in Figure 1, current SOTA offline RL algorithms, such as EDAC and MSG, are vulnerable to various types of data corruption, with a particular susceptibility to observation and dynamics attacks. In contrast, IQL exhibits enhanced resilience to observation, action, and reward attacks, although it still struggles with dynamic attack. Remarkably, Behavior Cloning (BC) undergoes a mere $0.4\%$ average performance degradation under random corruption. However, BC sacrifices performance without considering future returns. Trading off between performance and robustness, weighted imitation learning, to which IQL belongs, is thus more favorable. In Appendix E.1, we perform ablation studies on IQL and uncover evidence indicating that the supervised policy learning scheme contributes more to IQL's robustness than other techniques, such as expectile regression. This finding motivates us to theoretically understand such a learning scheme.

## 4.2 THEORETICAL ANALYSIS

We define the cumulative corruption level as follows.

**Assumption 1** (Cumulative Corruption). *Let $\zeta = \sum_{i=1}^{N}(2\zeta_i + \log \zeta_i')$ denote the cumulative corruption level, where $\zeta_i$ and $\zeta_i'$ are defined as*

$$\|[\mathcal{T}V](s_i, a) - [\tilde{\mathcal{T}}V](s_i, a)\|_\infty \leq \zeta_i, \quad \max\left\{\frac{\pi_{\mathcal{D}}(a \mid s_i)}{\pi_\mu(a \mid s_i)}, \frac{\pi_\mu(a \mid s_i)}{\pi_{\mathcal{D}}(a \mid s_i)}\right\} \leq \zeta_i', \quad \forall a \in A.$$

*Here $\|\cdot\|_\infty$ means taking supremum over $V : S \mapsto [0, R_{\max}/(1-\gamma)]$.*

We remark that $\{\zeta_i\}_{i=1}^{N}$ in Assumption 1 quantifies the corruption level of reward functions and transition dynamics, and $\{\zeta_i'\}_{i=1}^{N}$ represents the corruption level of observations and actions. Although $\{\zeta_i'\}_{i=1}^{N}$ could be relatively large, our final results only have a logarithmic dependency on them. In particular, if there is no observation and action corruptions, $\log \zeta_i' = \log 1 = 0$ for any $i = 1, \cdots, N$.

To facilitate our theoretical analysis, we assume that we have obtained the optimal value function $V^*$ by solving (1) and (2). This has been demonstrated in Kostrikov et al. (2021) under the uncorrupted setting. We provide justification for this under the corrupted setting in the discussion part of Appendix C.2. Then the policy update rule in (3) takes the following form:

$$\pi_{\text{IQL}} = \arg\max_\pi \mathbb{E}_{(s,a)\sim\mu}[\exp(\beta \cdot [\mathcal{T}V^* - V^*](s,a)) \log \pi(a \mid s)]$$
$$= \arg\min_\pi \mathbb{E}_{s\sim\mu}[\text{KL}(\pi_{\text{E}}(\cdot \mid s), \pi(\cdot \mid s))], \tag{5}$$

where $\pi_{\text{E}}(a \mid s) \propto \pi_\mu(a \mid s) \cdot \exp(\beta \cdot [\mathcal{T}V^* - V^*](s,a))$ is the policy that IQL imitates. However, the learner only has access to the corrupted dataset $\mathcal{D}$. With $\mathcal{D}$, IQL finds the following policy

$$\tilde{\pi}_{\text{IQL}} = \arg\min_\pi \mathbb{E}_{s\sim\mathcal{D}}[\text{KL}(\tilde{\pi}_{\text{E}}(\cdot \mid s), \pi(\cdot \mid s))], \tag{6}$$

where $\tilde{\pi}_{\text{E}}(a \mid s) \propto \pi_{\mathcal{D}}(a \mid s) \cdot \exp(\beta \cdot [\tilde{\mathcal{T}}V^* - V^*](s,a))$.

To bound the value difference between $\pi_{\text{IQL}}$ and $\tilde{\pi}_{\text{IQL}}$, our result relies on the coverage assumption.

**Assumption 2** (Coverage). *There exists an $M > 0$ satisfying $\max\{\frac{d^{\pi_{\text{E}}}(s,a)}{\mu(s,a)}, \frac{d^{\tilde{\pi}_{\text{E}}}(s,a)}{d_{\mathcal{D}}(s,a)}\} \leq M$ for any $(s,a) \in S \times A$.*

Assumption 2 requires that the corrupted dataset $\mathcal{D}$ and the clean data $\mu$ have good coverage of $\tilde{\pi}_{\text{E}}$ and $\pi_{\text{E}}$ respectively, similar to the partial coverage condition in (Jin et al., 2021; Rashidinejad et al., 2021). The following theorem shows that IQL is robust to corrupted data. For simplicity, we choose $\beta = 1$ in (5) and (6). Our analysis is ready to be extended to the case where $\beta$ takes on any constant.

**Theorem 3** (Robustness of IQL). *Fix $\beta = 1$. Under Assumptions 1 and 2, it holds that*

$$V^{\pi_{\text{IQL}}} - V^{\tilde{\pi}_{\text{IQL}}} \leq \frac{\sqrt{2M}R_{\max}}{(1-\gamma)^2}[\sqrt{\epsilon_1} + \sqrt{\epsilon_2}] + \frac{2R_{\max}}{(1-\gamma)^2}\sqrt{\frac{M\zeta}{N}},$$

*where $\epsilon_1 = \mathbb{E}_{s\sim\mu}[\text{KL}(\pi_{\text{E}}(\cdot \mid s), \pi_{\text{IQL}}(\cdot \mid s))]$ and $\epsilon_2 = \mathbb{E}_{s\sim\mathcal{D}}[\text{KL}(\tilde{\pi}_{\text{E}}(\cdot \mid s), \tilde{\pi}_{\text{IQL}}(\cdot \mid s))]$.*

See Appendix C.1 for a detailed proof. The $\epsilon_1$ and $\epsilon_2$ are standard imitation errors under clean data and corrupted data, respectively. These imitation errors can also be regarded as the generalization error of supervised learning objectives (5) and (6). As $N$ increases, this type of error typically diminishes (Janner et al., 2019; Xu et al., 2020). Besides, the corruption error term decays to zero with the mild assumption that $\zeta = o(N)$. Combining these two facts, we can conclude that IQL exhibits robustness in the face of diverse data corruption scenarios. Furthermore, we draw a comparison with a prior theoretical work (Zhang et al., 2022) in robust offline RL (refer to Remark 7 in Appendix C.1).

## 5 ROBUST IQL FOR DIVERSE DATA CORRUPTION

Our theoretical analysis suggests that the adoption of weighted imitation learning inherently offers robustness under data corruption. However, empirical results indicate that IQL still remains susceptible to dynamics attack. To enhance its resilience against all forms of data corruption, we introduce three improvements below.

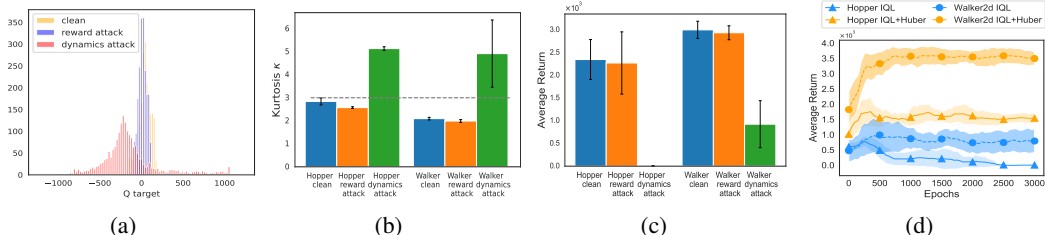

| (a) | (b) | (c) | (d) |

Figure 3: (a) Heavy tailedness of Q target distributions on the Hopper task. The target values are normalized by subtracting the mean. (b) Kurtosis values of Q target distribution on Hopper and Walker tasks. The gray dashed line represents the kurtosis value of a Gaussian distribution. (c) Average final returns of IQL with different datasets. (d) Comparison of IQL w/ and w/o Huber loss.

## 5.1 OBSERVATION NORMALIZATION

In the context of offline RL, previous studies have suggested that normalization does not yield significant improvements (Fujimoto & Gu, 2021). However, we find that normalization is highly advantageous in data corruption settings. Normalization is likely to help ensure that the algorithm's performance is not unduly affected by features with large-scale values. As depicted in Figure 2, it is evident that normalization enhances the performance of IQL across diverse attacks. More details regarding the ablation experiments on normalization is deferred to Appendix E.2. Given that $s$ and $s'$ in $(s, a, r, s')$ can be corrupted independently, the normalization is conducted by calculating the mean $\mu_o$ and variance $\sigma_o$ of all states and next-states in $\mathcal{D}$. Based on $\mu_o$ and $\sigma_o$, the states and next-states are normalized as $s_i = \frac{(s_i - \mu_o)}{\sigma_o}, s'_i = \frac{(s'_i - \mu_o)}{\sigma_o}$, where $\mu_o = \frac{1}{2N} \sum_{i=1}^{N} (s_i + s'_i)$ and $\sigma_o^2 = \frac{1}{2N} \sum_{i=1}^{N} [(s_i - \mu_o)^2 + (s'_i - \mu_o)^2]$.

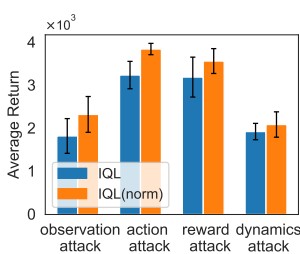

Figure 2: Average performance of IQL w/ and w/o normalization over four datasets.

## 5.2 HUBER LOSS FOR ROBUST VALUE FUNCTION LEARNING

As noted in Section 4.1, IQL, along with other offline RL algorithms, is susceptible to dynamics attack. The question arises: why are dynamics attacks challenging to handle? While numerous influential factors may contribute, we have identified an intriguing phenomenon: dynamics corruption introduces heavy-tailed targets for learning the Q function, which significantly impacts the performance of IQL. In Figure 3 (a), we visualize the distribution of the Q target, i.e., $r(s, a) + \gamma V_\psi(s')$, which is essentially influenced by the rewards and the transition dynamics. Additionally, we collected 2048 samples to calculate statistical values after training IQL for $3 \times 10^6$ steps. As depicted in Figure 3 (a), the reward attack does not significantly impact the target distribution, whereas the dynamics attack markedly reshapes the target distribution into a heavy-tailed one. In Figure 3 (b), we employ the metric Kurtosis $\kappa = \frac{\sum_{i=1}^{N} (X_i - \bar{X})^4 / N}{\sum_{i=1}^{N} ((X_i - \bar{X})^2 / N)^2}$, which measures the heavy-tailedness relative to a normal distribution (Mardia, 1970; Garg et al., 2021). The results demonstrate that dynamics corruption significantly increases the Kurtosis value compared to the clean dataset and the reward-attacked datasets. Correspondingly, as shown in Figure 3 (c), a higher degree of heavy-tailedness generally correlates with significantly lower performance. This aligns with our theory, as the heavy-tailed issue leads to a substantial increase in $\{\zeta_i\}_{i=1}^{N}$, making the upper bound in Theorem 3 very loose. This observation underscores the challenges posed by dynamics corruption and the need for robust strategies to mitigate its impact.

The above finding motivates us to utilize Huber loss (Huber, 1973; 1992), which is commonly employed to mitigate the issue of heavy-tailedness (Sun et al., 2020; Roy et al., 2021). In specific, given $V_\psi$, we replace the quadratic loss in (1) by Huber loss $l_H^\delta(\cdot)$ and obtain the following objective

$$\mathcal{L}_Q = \mathbb{E}_{(s,a,r,s') \sim \mathcal{D}} \left[ l_H^\delta (r + \gamma V(s') - Q(s, a)) \right], \quad \text{where } l_H^\delta(x) = \begin{cases} \frac{1}{2\delta} x^2, & \text{if } |x| \leq \delta \\ |x| - \frac{1}{2}\delta, & \text{if } |x| > \delta \end{cases}. \quad (7)$$

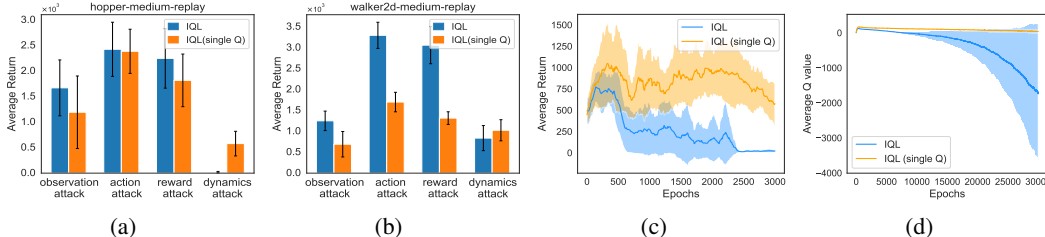

Figure 4: Comparison of IQL and IQL with single Q function in the (a) Hopper and (b) Walker tasks. (c) Average returns and (d) average Q values in the Hopper task under dynamics attack. IQL can be unstable due to the minimum operator.

Here $\delta$ trades off the resilience to outliers from $\ell_1$ loss and the rapid convergence of $\ell_2$ loss. The $\ell_1$ loss can result in a geometric median estimator that is robust to outliers and heavy-tailed distributions. As illustrated in Figure 3 (d), upon integrating Huber loss as in (7), we observe a significant enhancement in the robustness against dynamic attack. Next, we provide a theoretical justification for the usage of Huber loss. We first adopt the following heavy-tailed assumption.

**Assumption 4.** *Fix the value function parameter $\psi$, for any $(s, a, r, s') \sim \mathcal{D}$, it holds that $r + \gamma V_\psi(s') = r(s, a) + \gamma \mathbb{E}_{\tilde{s} \sim P(\cdot|s,a)}[V_\psi(\tilde{s})] + \varepsilon_{s,a,s'}$, where $\varepsilon_{s,a,s'}$ is the heavy-tailed noise satisfying $\mathbb{E}[\varepsilon_{s,a,s'} \mid s, a, s'] = 0$ and $\mathbb{E}[|\varepsilon_{s,a,s'}|^{1+\nu}] < \infty$ for some $\nu > 0$.*

Assumption 4 only assumes the existence of the $(1 + \nu)$-th moment of the noise $\varepsilon_{s,a,s'}$, which is strictly weaker than the standard boundedness or sub-Gaussian assumption. This heavy-tailed assumption is widely used in heavy-tailed bandits (Bubeck et al., 2013; Shao et al., 2018) and heavy-tailed RL (Zhuang & Sui, 2021; Huang et al., 2023). Notably, the second moment of $\varepsilon_{s,a,s'}$ may not exist when $\nu < 1$, leading to the square loss inapplicable in the heavy-tailed setting. In contrast, by adopting the Huber regression, we obtain the following theoretical guarantee:

**Lemma 5.** *Under Assumption 4, if we solve the Huber regression problem in* (7) *and obtain the estimated Q-function $\hat{Q}$, then we have*

$$\|\hat{Q}(s, a) - r(s, a) - \gamma \mathbb{E}_{s' \sim P(\cdot|s,a)}[V_\psi(s')]\|_\infty \lesssim N^{-\min\{\nu/(1+\nu), 1/2\}},$$

*where $\|\cdot\|_\infty$ means taking supremum over $(s, a) \in S \times A$ and $N$ is the size of dataset $\mathcal{D}$. Furthermore, if $N \to \infty$, we have $\hat{Q}(s, a) \to r(s, a) + \gamma \mathbb{E}_{s' \sim P(\cdot|s,a)}[V_\psi(s')]$ for any $(s, a) \in S \times A$.*

The detailed proof is deferred to Appendix C.2. Thus, by utilizing the Huber loss, we can recover the nearly unbiased value function even in the presence of a heavy-tailed target distribution.

### 5.3 PENALIZING CORRUPTED DATA VIA IN-DATASET UNCERTAINTY

We have noted a key technique within IQL's implementation - the Clipped Double Q-learning (CDQ) (Fujimoto et al., 2018). CDQ learns two Q functions and utilizes the minimum one for both value and policy updating. To verify its effectiveness, we set another variant of IQL without the CDQ trick, namely "IQL(single Q)", which only learns a single Q function for IQL updates. As shown in Figure 4 (a) and (b), "IQL(single Q)" notably underperforms IQL on most corruption settings except the dynamics corruption. The intrigue lies within the minimum operator's correlation to the lower confidence bound (LCB) (An et al., 2021). Essentially, it penalizes corrupted data, as corrupted data typically exhibits greater uncertainty, consequently reducing the influence of these data points on the policy. Further results in Figure 4 (c) and (d) explain why CDQ decreases the performance under dynamics corruption. Since the dynamics corruption introduces heavy-tailedness in the Q target as explained in Section 5.2, the minimum operator would persistently reduce the value estimation, leading to a negative value exploding. Instead, "IQL(single Q)" learns an average Q value, as opposed to the minimum operator, resulting in better performance under dynamics corruption.

The above observation inspires us to use the CDQ trick safely by extending it to quantile Q estimators using an ensemble of Q functions $\{Q_{\theta_i}\}_{i=1}^K$. The $\alpha$-quantile value $Q_\alpha$ is defined as $\Pr[Q < Q_\alpha] = \alpha$, where $\alpha \in [0, 1]$. A detailed calculation of the quantiles is deferred to the

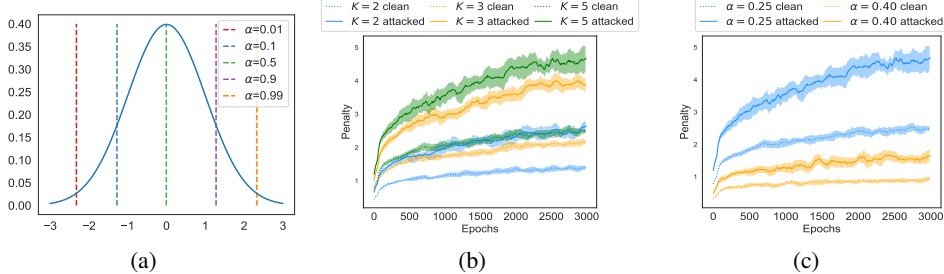

Figure 5: (a) Quantiles of a normal distribution. In-dataset penalty for attacked data and clean data across (b) different number of $K$ and (c) different quantile value $\alpha$.

Appendix D.3. The quantiles of a normal distribution are depicted in Figure 5 (a). When $\alpha$ corresponds to an integer index and the Q value follows a Gaussian distribution with mean $m(s, a)$ and variance $\sigma^2(s, a)$, it is well-known (Blom, 1958; Royston, 1982) that the $\alpha$-quantile is correlated to the LCB estimation of a variable: $\mathbb{E}\left[Q_\alpha(s, a)\right] \approx m(s, a) - \Phi^{-1}\left(\frac{(1-\alpha) \times (K-1) + 1 - 0.375}{K - 2 \times 0.375 + 1}\right) \cdot \sigma(s, a)$.

Hence, we can control the degree of penalty by controlling $\alpha$ and the ensemble size $K$. In Figure 5 (b) and (c), we illustrate the penalty, defined as $m(s, a) - Q_\alpha(s, a)$, for both clean and corrupted parts of the dataset. It becomes evident that (1) the attacked portion of data can incur a heavier penalty compared to the clean part, primarily due to the larger uncertainty and a higher estimation variance $\sigma^2(s, a)$ associated with these data points, and (2) the degree of penalty can be adjusted flexibly to be larger with a larger $K$ or a smaller $\alpha$. Notably, when $K = 2$ and $\alpha = 0$, the quantile Q estimator recovers the CDQ trick in IQL. We will discuss the impact of the quantile Q estimator on the equations of RIQL in Section 5.4.

## 5.4 RIQL Algorithm

We summarize the overall RIQL algorithm, whose pseudocode is provided in Algorithm 1 (Appendix B). First, RIQL normalizes the states and next states according to Section 5.1. The ensemble Q value functions $\{Q_{\theta_i}(s, a)\}_{i=1}^K$ are learned by solving the Huber regression

$$\mathcal{L}_Q(\theta_i) = \mathbb{E}_{(s,a,r,s')\sim\mathcal{D}}[l_H^\delta(r + \gamma V_\psi(s') - Q_{\theta_i}(s, a))], \tag{8}$$

where $l_H^\delta(\cdot)$ is the Huber loss defined in (7). The value function $V_\psi(s)$ is learned to minimize

$$\mathcal{L}_V(\psi) = \mathbb{E}_{(s,a)\sim\mathcal{D}}\left[\mathcal{L}_2^\tau(Q_\alpha(s, a) - V_\psi(s))\right], \quad \mathcal{L}_2^\tau(x) = |\tau - \mathbf{1}(x < 0)|x^2. \tag{9}$$

Here $Q_\alpha(s, a)$ is defined by the $\alpha$-quantile value among $\{Q_{\theta_i}(s, a)\}_{i=1}^K$. The policy is learned to maximize the $\alpha$-quantile advantage-weighted imitation learning objective:

$$\mathcal{L}_\pi(\phi) = \mathbb{E}_{(s,a)\sim\mathcal{D}}\left[\exp(\beta A_\alpha(s, a)) \log \pi_\phi(a|s)\right], \tag{10}$$

where the advantage $A_\alpha(s, a) = Q_\alpha(s, a) - V_\psi(s)$. Intuitively, the quantile Q estimator serves a dual purpose: (1) underestimating the Q values with high uncertainty in (9), and (2) downweighting these samples to reduce their impact on the policy in (10). An essential difference between our quantile Q estimator and those in prior works (An et al., 2021; Bai et al., 2022; Ghasemipour et al., 2022) is that we penalize **in-dataset data with high uncertainty** due to the presence of corrupted data, whereas conventional offline RL leverages uncertainty to penalize **out-of-distribution actions**.

## 6 Experiments

In this section, we empirically evaluate the performance of RIQL across diverse data corruption scenarios, including both random and adversarial attacks on states, actions, rewards, and next-states.

**Experimental Setup.** Two parameters, corruption rate $c \in [0, 1]$ and corruption scale $\epsilon$ are used to control the cumulative corruption level. **Random corruption** is applied by adding random noise to the attacked elements of a $c$ portion of the datasets. In contrast, **adversarial corruption** is introduced through Projected Gradient Descent attack (Madry et al., 2017; Zhang et al., 2020) with

Table 1: Average normalized performance under random data corruption.

| Environment | Attack Element | BC | EDAC | MSG | CQL | SQL | IQL | RIQL (ours) |
|---|---|---|---|---|---|---|---|---|
| Halfcheetah | observation | **33.4±1.8** | 2.1±0.5 | -0.2±2.2 | 9.0±7.5 | 4.1±1.4 | 21.4±1.9 | 27.3±2.4 |
| | action | 36.2±0.3 | 47.4±1.3 | **52.0±0.9** | 19.9±21.3 | 42.9±0.4 | 42.2±1.9 | 42.9±0.6 |
| | reward | 35.8±0.9 | 38.6±0.3 | 17.5±16.4 | 32.6±19.6 | 41.7±0.8 | 42.3±0.4 | **43.6±0.6** |
| | dynamics | 35.8±0.9 | 1.5±0.2 | 1.7±0.4 | 29.2±4.0 | 35.5±0.4 | 36.7±1.8 | **43.1±0.2** |
| Walker2d | observation | 9.6±3.9 | -0.2±0.3 | -0.4±0.1 | 19.4±1.6 | 0.6±1.0 | 27.2±5.1 | **28.4±7.7** |
| | action | 18.1±2.1 | 83.2±1.9 | 25.3±10.6 | 62.7±7.2 | 76.0±4.2 | 71.3±7.8 | **84.6±3.3** |
| | reward | 16.0±7.4 | 4.3±3.6 | 18.4±9.5 | 69.4±7.4 | 33.8±13.8 | 65.3±8.4 | **83.2±2.6** |
| | dynamics | 16.0±7.4 | -0.1±0.0 | 7.4±3.7 | -0.2±0.1 | 15.3±2.2 | 17.7±7.3 | **78.2±1.8** |
| Hopper | observation | 21.5±2.9 | 1.0±0.5 | 6.9±5.0 | 42.8±7.0 | 5.2±1.9 | 52.0±16.6 | **62.4±1.8** |
| | action | 22.8±7.0 | **100.8±0.5** | 37.6±6.5 | 69.8±4.5 | 73.4±7.3 | 76.3±15.4 | 90.6±5.6 |
| | reward | 19.5±3.4 | 2.6±0.7 | 24.9±4.3 | 70.8±8.9 | 52.3±1.7 | 69.7±18.8 | **84.8±13.1** |
| | dynamics | 19.5±3.4 | 0.8±0.0 | 12.4±4.9 | 0.8±0.0 | 24.3±5.6 | 1.3±0.5 | **51.5±8.1** |
| Average score ↑ | | 23.7 | 23.5 | 17.0 | 35.5 | 33.8 | 43.6 | **60.0** |
| Average degradation percentage ↓ | | 0.4% | 68.5% | 61.5% | 42.3% | 45.0% | 31.2% | 17.0% |

the pretrained value functions. A unique case is the adversarial reward corruption, which directly multiplies $-\epsilon$ to the original rewards instead of performing gradient optimization. We utilize the "medium-replay-v2" dataset from (Fu et al., 2020), which better represents real scenarios because it is collected during the training of a SAC agent. The corruption rate is set to $c = 0.3$ and the corruption scale is $\epsilon = 1$ for the main results. More details about all forms of data corruption used in our experiments are provided in Appendix D.1. We compare RIQL with SOTA offline RL algorithms, including ensemble-based algorithms such as EDAC (An et al., 2021) and MSG (Ghasemipour et al., 2022), as well as ensemble-free baselines like IQL (Kostrikov et al., 2021), CQL (Kumar et al., 2020), and SQL (Xu et al., 2023). To evaluate the performance, we test each trained agent in a clean environment and average their performance over four random seeds. Implementation details and additional empirical results are provided in Appendix D and Appendix E, respectively.

## 6.1 EVALUATION UNDER RANDOM AND ADVERSARIAL CORRUPTION

**Random Corruption**  We begin by evaluating various algorithms under random data corruption. Table 1 demonstrates the average normalized performance (calculated as $100 \times \frac{\text{score} - \text{random score}}{\text{expert score} - \text{random score}}$) of different algorithms. Overall, RIQL demonstrates superior performance compared to other baselines, achieving an average score improvement of 37.6% over IQL. In 9 out of 12 settings, RIQL achieves the highest results, particularly excelling in the dynamics corruption settings, where it outperforms other methods by a large margin. Among the baselines, it is noteworthy that BC experiences the least average degradation percentage compared to its score on the clean dataset. This can be attributed to its supervised learning scheme, which avoids incorporating future information but also results in a relatively low score. Besides, offline RL methods like EDAC and MSG exhibit performance degradation exceeding 60%, highlighting their unreliability under data corruption.

**Adversarial Corruption**  To further investigate the robustness of RIQL under a more challenging corruption setting, we consider an adversarial corruption setting. As shown in Table 2, RIQL consistently surpasses other baselines by a significant margin. Notably, RIQL improves the average score by almost 70% over IQL. When compared to the results of random corruption, every algorithm experiences a larger performance drop. All baselines, except for BC, undergo a performance drop of more than or close to half. In contrast, RIQL's performance only diminishes by 22% compared to its score on the clean data. This confirms that adversarial attacks present a more challenging setting and further highlights the effectiveness of RIQL against various data corruption scenarios.

## 6.2 EVALUATION WITH VARYING CORRUPTION RATES

In the above experiments, we employ a constant corruption rate of 0.3. In this part, we assess RIQL and other baselines under varying corruption rates ranging from 0.0 to 0.5. The results are depicted in Figure 6. RIQL exhibits the most robust performance compared to other baselines, particularly under reward and dynamics attacks, due to its effective management of heavy-tailed targets using the Huber loss. Notably, EDAC and MSG are highly sensitive to a minimal corruption rate of 0.1 under observation, reward, and dynamics attacks. Furthermore, among the attacks on the four elements, the observation attack presents the greatest challenge for RIQL. This can be attributed to

Table 2: Average normalized performance under adversarial data corruption.

| Environment | Attack Element | BC | EDAC | MSG | CQL | SQL | IQL | RIQL (ours) |
|---|---|---|---|---|---|---|---|---|
| Halfcheetah | observation | 34.5±1.5 | 1.1±0.3 | 1.1±0.2 | 5.0±11.6 | 8.3±0.9 | 32.6±2.7 | **35.7±4.2** |
| | action | 14.0±1.1 | 32.7±0.7 | **37.3±0.7** | -2.3±1.2 | 32.7±1.0 | 27.5±0.3 | 31.7±1.7 |
| | reward | 35.8±0.9 | 40.3±0.5 | **47.7±0.4** | -1.7±0.3 | 42.9±0.1 | 42.6±0.4 | 44.1±0.8 |
| | dynamics | 35.8±0.9 | -1.3±0.1 | -1.5±0.0 | -1.6±0.0 | 10.4±2.6 | 26.7±0.7 | **35.8±2.1** |
| Walker2d | observation | 12.7±5.9 | -0.0±0.1 | 2.9±2.7 | 61.8±7.4 | 1.8±1.9 | 37.7±13.0 | **70.0±5.3** |
| | action | 5.4±0.4 | 41.9±24.0 | 5.4±0.9 | 27.0±7.5 | 31.3±8.8 | 27.5±0.6 | **66.1±4.6** |
| | reward | 16.0±7.4 | 57.3±33.2 | 9.6±4.9 | 67.0±6.1 | 78.1±2.0 | 73.5±4.85 | **85.0±1.5** |
| | dynamics | 16.0±7.4 | 4.3±0.9 | 0.1±0.2 | 3.9±1.4 | 2.7±1.9 | -0.1±0.1 | **60.6±21.8** |
| Hopper | observation | 21.6±7.1 | 36.2±16.2 | 16.0±2.8 | **78.0±6.5** | 8.2±4.7 | 32.8±6.4 | 50.8±7.6 |
| | action | 15.5±2.2 | 25.7±3.8 | 23.0±2.1 | 32.2±7.6 | 30.0±0.4 | 37.9±4.8 | **63.6±7.3** |
| | reward | 19.5±3.4 | 21.2±1.9 | 22.6±2.8 | 49.6±12.3 | 57.9±4.8 | 57.3±9.7 | **65.8±9.8** |
| | dynamics | 19.5±3.4 | 0.6±0.0 | 0.6±0.0 | 0.6±0.0 | 18.9±12.6 | 1.3±1.1 | **65.7±21.1** |
| Average score ↑ | | 20.5 | 21.7 | 13.7 | 26.6 | 25.8 | 33.1 | **56.2** |
| Average degradation percentage ↓ | | 13.4% | 71.2% | 69.9% | 66.8% | 57.5% | 46.0% | 22.0% |

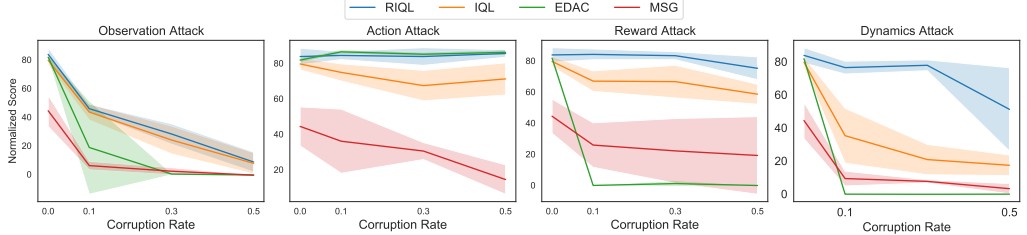

Figure 6: Varying the corruption rate of random attacks on the Walker2d task.

the additional distribution shift introduced by the observation corruption, highlighting the need for further enhancements.

### 6.3 ABLATIONS

We assess the individual contributions of RIQL's components under the mixed attack scenario, where all four elements are corrupted independently, each with a corruption rate of 0.2 and a corruption range of 1.0. We consider the following RIQL variants: RIQL without observation normalization (**RIQL w/o norm**), RIQL without the quantile Q estimator (**RIQL w/o quantile**), and RIQL without the Huber loss (**RIQL w/o Huber**). The average results for the Walker and Hopper tasks on the "medium-replay-v2" dataset are presented in Figure 7. These results indicate that each component contributes to RIQL's performance, with the Huber loss standing out as the most impactful factor due to its effectiveness in handling the dynamics corruption. The best performance is achieved when combining all three components into RIQL.

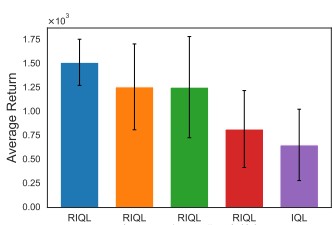

Figure 7: Ablation of RIQL under mixed attack.

### 7 CONCLUSION

In this paper, we investigate the robustness of offline RL algorithms in the presence of diverse data corruption, including states, actions, rewards, and dynamics. Our empirical observations reveal that current offline RL algorithms are notably vulnerable to different forms of data corruption, particularly dynamics corruption, posing challenges for their application in real-world scenarios. To tackle this issue, we introduce Robust IQL (RIQL), an offline RL algorithm that incorporates three simple yet impactful enhancements: observation normalization, Huber loss, and quantile Q estimators. Through empirical evaluations conducted under both random and adversarial attacks, we demonstrate RIQL's exceptional robustness against various types of data corruption. We hope that our work will inspire further research aimed at addressing data corruption in more realistic scenarios.

ACKNOWLEDGMENTS

The work is done when Rui Yang and Tong Zhang were at HKUST. The authors would like to thank Chenlu Ye for insightful discussions and the anonymous reviewers for their valuable feedback.

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

# A  RELATED WORKS

**Offline RL.** Ensuring that the policy distribution remains close to the data distribution is crucial for offline RL, as distributional shift can lead to unreliable estimation (Levine et al., 2020). Offline RL algorithms typically fall into two categories to address this: those that enforce policy constraints on the learned policy (Wang et al., 2018; Fujimoto et al., 2019; Peng et al., 2019; Li et al., 2020; Fujimoto & Gu, 2021; Kostrikov et al., 2021; Chen et al., 2021; Emmons et al., 2021; Sun et al., 2024; Xu et al., 2023), and those that learn pessimistic value functions to penalize OOD actions (Kumar et al., 2020; Yu et al., 2020; An et al., 2021; Bai et al., 2022; Yang et al., 2022a; Ghasemipour et al., 2022; Sun et al., 2022a; Nikulin et al., 2023). Among these algorithms, ensemble-based approaches (An et al., 2021; Ghasemipour et al., 2022) demonstrate superior performance by estimating the lower-confidence bound (LCB) of Q values for OOD actions based on uncertainty estimation (Loquercio et al., 2020; Sun et al., 2022b). Additionally, imitation-based (Emmons et al., 2021) or weighted imitation-based algorithms (Wang et al., 2018; Nair et al., 2020; Kostrikov et al., 2021) generally enjoy better simplicity and stability compared to other pessimism-based approaches. Studies in offline goal-conditioned RL (Yang et al., 2023; 2022b) further demonstrate that weighted imitation learning offers improved generalization compared to pessimism-based offline RL methods. In addition to traditional offline RL methods, recent studies employ advanced techniques such as transformer (Chen et al., 2021; Chebotar et al., 2023; Yamagata et al., 2023) and diffusion models (Janner et al., 2022; Hansen-Estruch et al., 2023; Wang et al., 2022) for sequence modeling or function class enhancement, thereby increasing the potential of offline RL to tackle more challenging tasks.

**Offline RL Theory.** There is a large body of literature (Jin et al., 2021; Rashidinejad et al., 2021; Zanette et al., 2021; Xie et al., 2021a;b; Uehara & Sun, 2021; Shi et al., 2022; Zhong et al., 2022; Cui & Du, 2022; Xiong et al., 2022; Li et al., 2022; Cheng et al., 2022) dedicating to the development of pessimism-based algorithms for offline RL. These algorithms can provably efficiently find near-optimal policies with only the partial coverage condition. However, these works do not consider the corrupted data. Moreover, when the cumulative corruption level is sublinear, i.e., $\zeta = o(N)$, our algorithm can handle the corrupted data under similar partial coverage assumptions.

**Robust RL.** One type of robust RL is the distributionally robust RL, which aims to learn a policy that optimizes the worst-case performance across MDPs within an uncertainty set, typically framed as a Robust MDP problem (Nilim & Ghaoui, 2003; Iyengar, 2005; Ho et al., 2018; Moos et al., 2022). Hu et al. (2022) prove that distributionally robust RL can effectively reduce the sim-to-real gap. Besides, numerous studies in the online setting have explored robustness to perturbations on observations (Zhang et al., 2020; 2021), actions (Pinto et al., 2017; Tessler et al., 2019), rewards (Wang et al., 2020; Husain et al., 2021), and dynamics (Mankowitz et al., 2019). There is also a line of theory works (Lykouris et al., 2021; Wu et al., 2021; Wei et al., 2022; Ye et al., 2023a) studying online corruption-robust RL. In the offline setting, a number of works focus on testing-time robustness or distributional robustness in offline RL (Zhou et al., 2021; Shi & Chi, 2022; Hu et al., 2022; Yang et al., 2022a; Panaganti et al., 2022; Wen et al., 2023; Blanchet et al., 2023). Regarding the training-time robustness of offline RL, Li et al. (2023) investigated reward attacks in offline RL, revealing that certain dataset biases can implicitly enhance offline RL's resilience to reward corruption. Wu et al. (2022) propose a certification framework designed to ascertain the number of tolerable poisoning trajectories in relation to various certification criteria. From a theoretical perspective, Zhang et al. (2022) studied offline RL under contaminated data. One concurrent work (Ye et al., 2023b) leverages uncertainty weighting to tackle reward and dynamics corruption with theoretical guarantees. Different from prior work, we propose an algorithm that is both provable and practical under diverse data corruption on all elements.

**Robust Imitation Learning.** Robust imitation learning focuses on imitating the expert policy using corrupted demonstrations (Liu et al., 2022) or a mixture of expert and non-expert demonstrations (Wu et al., 2019; Tangkaratt et al., 2020b;a; Sasaki & Yamashina, 2020). These approaches primarily concentrate on noise or attacks on states and actions, without considering the future return. In contrast, robust offline RL faces the intricate challenges associated with corruption in rewards and dynamics.

**Heavy-tailedness in RL.** In the realm of RL, Zhuang & Sui (2021); Huang et al. (2023) delved into the issue of heavy-tailed rewards in tabular Markov Decision Processes (MDPs) and function approximation, respectively. There is also a line of works (Bubeck et al., 2013; Shao et al., 2018;

Xue et al., 2020; Zhong et al., 2021; Huang et al., 2022; Kang & Kim, 2023) studying the heavy-tailed bandit, which is a special case of MDPs. Besides, Garg et al. (2021) investigated the heavy-tailed gradients in the training of Proximal Policy Optimization. In contrast, our work addresses the heavy-tailed target distribution that emerges from data corruption.

**Huber Loss in RL.** The Huber loss, known for its robustness to outliers, has been widely employed in the Deep Q-Network (DQN) literature, (Dabney et al., 2018; Agarwal et al., 2020; Patterson et al., 2022). However, Ceron & Castro (2021) reevaluated the Huber loss and discovered that it fails to outperform the MSE loss on MinAtar environments. In our study, we leverage the Huber loss to address the heavy-tailedness in Q targets caused by data corruption, and we demonstrate its remarkable effectiveness.

## B   ALGORITHM PSEUDOCODE

The pseudocode of RIQL and IQL can be found in Algorithm 1 and Algorithm 2, respectively:

---

**Algorithm 1:** Robust IQL algorithm

---

Initialize policy $\pi_\phi$ and value function $V_\psi, Q_{\theta_i}, i \in [1, K]$ ;
Normalize the observation according to Section 5.1 ;
**for** *training step*$= 1, 2, \ldots, T$ **do**

 Sample a mini-batch from the offline dataset: $\{(s, a, r, s')\} \sim D$;
 Update value function $V_\psi$ to minimize

$$\mathcal{L}_V(\psi) = \mathbb{E}_{(s,a) \sim \mathcal{D}} \left[ \mathcal{L}_2^\tau (Q_\alpha(s, a) - V_\psi(s)) \right] ;$$

 Update $\{Q_{\theta_i}\}_{i=1}^K$ independently to minimize

$$\mathcal{L}_Q(\theta_i) = \mathbb{E}_{(s,a,r,s') \sim \mathcal{D}} [l_H^\delta (r + \gamma V_\psi(s') - Q_{\theta_i}(s, a))];$$

 Update policy $\pi_\phi$ to maximize

$$\mathcal{L}_\pi(\phi) = \mathbb{E}_{(s,a) \sim \mathcal{D}} \left[ \exp(\beta A_\alpha(s, a)) \log \pi_\phi(a|s) \right]$$

**end for**

---

---

**Algorithm 2:** IQL algorithm (Kostrikov et al., 2021) (for comparison)

---

Initialize policy $\pi_\phi$ and value function $V_\psi, Q_\theta$ ;
**for** *training step*$= 1, 2, \ldots, T$ **do**

 Sample a mini-batch from the offline dataset: $\{(s, a, r, s')\} \sim D$;
 Update value function $V_\psi$ to minimize (2) ;
 Update $Q_\theta$ to minimize (1) ;
 Update policy $\pi_\phi$ to maximize (3)
**end for**

---

## C   THEORETICAL ANALYSIS

### C.1   PROOF OF THEOREM 3

Before giving the proof of Theorem 3, we first state the performance difference lemma.

**Lemma 6** (Performance Difference Lemma (Kakade & Langford, 2002))**.** *For any $\pi$ and $\pi'$, it holds that*

$$V^\pi - V^{\pi'} = \frac{1}{1-\gamma} \mathbb{E}_{s \sim d^\pi} \mathbb{E}_{a \sim \pi(\cdot|s)} [A^{\pi'}(s, a)].$$

*Proof.* See Kakade & Langford (2002) for a detailed proof. □

*Proof of Theorem 3.* First, we have

$$V^{\pi_{\mathrm{IQL}}} - V^{\tilde{\pi}_{\mathrm{IQL}}} = \underbrace{V^{\pi_{\mathrm{IQL}}} - V^{\pi_{\mathrm{E}}}}_{\text{imitation error 1}} + \underbrace{V^{\pi_{\mathrm{E}}} - V^{\tilde{\pi}_{\mathrm{E}}}}_{\text{corruption error}} + \underbrace{V^{\tilde{\pi}_{\mathrm{E}}} - V^{\tilde{\pi}_{\mathrm{IQL}}}}_{\text{imitation error 2}}.$$

We then analyze these three error terms respectively.

**Corruption Error.** By the performance difference lemma (Lemma 6), we have

$$
\begin{aligned}
V^{\pi_{\mathrm{E}}} - V^{\tilde{\pi}_{\mathrm{E}}} &= -\frac{1}{1-\gamma} \mathbb{E}_{s \sim d^{\tilde{\pi}_{\mathrm{E}}}} \mathbb{E}_{a \sim \tilde{\pi}_{\mathrm{E}}(\cdot|s)} [A^{\pi_{\mathrm{E}}}(s,a)] \\
&= \frac{1}{1-\gamma} \mathbb{E}_{s \sim d^{\tilde{\pi}_{\mathrm{E}}}} \mathbb{E}_{a \sim \tilde{\pi}_{\mathrm{E}}(\cdot|s)} [V^{\pi_{\mathrm{E}}}(s) - Q^{\pi_{\mathrm{E}}}(s,a)] \\
&= \frac{1}{1-\gamma} \mathbb{E}_{s \sim d^{\tilde{\pi}_{\mathrm{E}}}} \left[ \mathbb{E}_{a \sim \pi_{\mathrm{E}}(\cdot|s)} [Q^{\pi_{\mathrm{E}}}(s,a)] - \mathbb{E}_{a \sim \tilde{\pi}_{\mathrm{E}}(\cdot|s)} [Q^{\pi_{\mathrm{E}}}(s,a)] \right] \\
&\leq \frac{R_{\max}}{(1-\gamma)^2} \mathbb{E}_{s \sim d^{\tilde{\pi}_{\mathrm{E}}}} [\|\tilde{\pi}_{\mathrm{E}}(\cdot \mid s) - \pi_{\mathrm{E}}(\cdot \mid s)\|_1],
\end{aligned}
\tag{11}
$$

where the second inequality uses the fact that $A^{\pi_{\mathrm{E}}}(s,a) = Q^{\pi_{\mathrm{E}}}(s,a) - V^{\pi_{\mathrm{E}}}(s)$, and the last inequality is obtained by Hölder's inequality and the fact that $\|Q^{\pi_{\mathrm{E}}}\|_\infty \leq R_{\max}/(1-\gamma)$. By Pinsker's inequality, we further have

$$
\begin{aligned}
\mathbb{E}_{s \sim d^{\tilde{\pi}_{\mathrm{E}}}} [\|\tilde{\pi}_{\mathrm{E}}(\cdot \mid s) - \pi_{\mathrm{E}}(\cdot \mid s)\|_1] &\leq \mathbb{E}_{s \sim d^{\tilde{\pi}_{\mathrm{E}}}} \left[ \sqrt{2\mathrm{KL}\big(\tilde{\pi}_{\mathrm{E}}(\cdot \mid s), \pi_{\mathrm{E}}(\cdot \mid s)\big)} \right] \\
&\leq \sqrt{2\mathbb{E}_{(s,a) \sim d^{\tilde{\pi}_{\mathrm{E}}}} \log \frac{\tilde{\pi}_{\mathrm{E}}(a \mid s)}{\pi_{\mathrm{E}}(a \mid s)}} \\
&\leq \sqrt{2M\mathbb{E}_{(s,a) \sim d_{\mathcal{D}}} \log \frac{\tilde{\pi}_{\mathrm{E}}(a \mid s)}{\pi_{\mathrm{E}}(a \mid s)}} \\
&= \sqrt{\frac{2M}{N} \sum_{i=1}^{N} \log \frac{\tilde{\pi}_{\mathrm{E}}(a_i \mid s_i)}{\pi_{\mathrm{E}}(a_i \mid s_i)}},
\end{aligned}
\tag{12}
$$

where the second inequality follows from Jensen's inequality and the definition of KL-divergence, the third inequality is obtained by the coverage assumption (Assumption 2) that $\sup_{s,a} [d^{\tilde{\pi}_{\mathrm{E}}}(s,a)/d_{\mathcal{D}}(s,a)] \leq M$, and the last inequality uses the definition of $d_{\mathcal{D}}$. Recall that $\pi_{\mathrm{E}}$ and $\tilde{\pi}_{\mathrm{E}}$ take the following forms:

$$
\begin{aligned}
\pi_{\mathrm{E}}(a \mid s) &\propto \pi_\mu(a \mid s) \cdot \exp(\beta \cdot [\mathcal{T}V^* - V^*](s,a)), \\
\tilde{\pi}_{\mathrm{E}}(a \mid s) &\propto \pi_{\mathcal{D}}(a \mid s) \cdot \exp(\beta \cdot [\tilde{\mathcal{T}}V^* - V^*](s,a)).
\end{aligned}
$$

We have

$$
\begin{aligned}
\frac{\tilde{\pi}_{\mathrm{E}}(a_i \mid s_i)}{\pi_{\mathrm{E}}(a_i \mid s_i)} = &\frac{\pi_{\mathcal{D}}(a_i \mid s_i) \cdot \exp(\beta \cdot [\tilde{\mathcal{T}}V^* - V^*](s_i,a_i))}{\pi_\mu(a_i \mid s_i) \cdot \exp(\beta \cdot [\mathcal{T}V^* - V^*](s_i,a_i))} \\
&\times \frac{\sum_{a \in A} \pi_\mu(a \mid s_i) \cdot \exp(\beta \cdot [\mathcal{T}V^* - V^*](s_i,a))}{\sum_{a \in A} \pi_{\mathcal{D}}(a \mid s_i) \cdot \exp(\beta \cdot [\tilde{\mathcal{T}}V^* - V^*](s_i,a))}.
\end{aligned}
\tag{13}
$$

By the definition of corruption levels in Assumption 1, we have

$$\zeta_i = \|[\mathcal{T}V](s_i,a) - [\tilde{\mathcal{T}}V](s_i,a)\|_\infty, \quad \max\left\{ \frac{\pi_{\mathcal{D}}(a \mid s_i)}{\pi_\mu(a \mid s_i)}, \frac{\pi_\mu(a \mid s_i)}{\pi_{\mathcal{D}}(a \mid s_i)} \right\} \leq \zeta_i', \quad \forall a \in A.$$

This implies that

$$
\begin{aligned}
\frac{\pi_{\mathcal{D}}(a_i \mid s_i) \cdot \exp(\beta \cdot [\tilde{\mathcal{T}}V^* - V^*](s_i,a_i))}{\pi_\mu(a_i \mid s_i) \cdot \exp(\beta \cdot [\mathcal{T}V^* - V^*](s_i,a_i))} &\leq \zeta_i' \cdot \exp(\beta\zeta_i) \\
\frac{\pi_\mu(a \mid s_i) \cdot \exp(\beta \cdot [\mathcal{T}V^* - V^*](s_i,a))}{\pi_{\mathcal{D}}(a \mid s_i) \cdot \exp(\beta \cdot [\tilde{\mathcal{T}}V^* - V^*](s_i,a))} &\leq \zeta_i' \cdot \exp(\beta\zeta_i), \quad \forall a \in A.
\end{aligned}
\tag{14}
$$

Plugging (14) into (13), we have

$$
\begin{aligned}
\frac{\tilde{\pi}_{\mathrm{E}}(a_i \mid s_i)}{\pi_{\mathrm{E}}(a_i \mid s_i)} &\leq \zeta_i' \cdot \exp(\beta\zeta_i) \cdot \frac{\zeta_i' \cdot \exp(\beta\zeta_i) \sum_{a \in A} \pi_{\mathcal{D}}(a \mid s_i) \cdot \exp(\beta \cdot [\tilde{\mathcal{T}}V^* - V^*](s_i, a))}{\sum_{a \in A} \pi_{\mathcal{D}}(a \mid s_i) \cdot \exp(\beta \cdot [\tilde{\mathcal{T}}V^* - V^*](s_i, a))} \\
&= \zeta_i'^2 \cdot \exp(2\beta\zeta_i).
\end{aligned}
\tag{15}
$$

Combining (11), (12), and (15), we have

$$
V^{\pi_{\mathrm{E}}} - V^{\tilde{\pi}_{\mathrm{E}}} \leq \frac{2R_{\max}}{(1-\gamma)^2} \sqrt{\frac{M}{N} \sum_{i=1}^{N} (\log \zeta_i' + 2\beta\zeta_i)} = \frac{2R_{\max}}{(1-\gamma)^2} \sqrt{\frac{M\zeta}{N}},
\tag{16}
$$

where the last equality uses $\beta = 1$ and the definition of $\zeta$ in Assumption 1.

**Imitation Error 1.** Following the derivation of (11), we have

$$
\begin{aligned}
V^{\pi_{\mathrm{IQL}}} - V^{\pi_{\mathrm{E}}} &\leq \frac{R_{\max}}{(1-\gamma)^2} \mathbb{E}_{s \sim d^{\pi_{\mathrm{E}}}} [\|\pi_{\mathrm{E}}(\cdot \mid s) - \pi_{\mathrm{IQL}}(\cdot \mid s)\|_1] \\
&\leq \frac{\sqrt{2}R_{\max}}{(1-\gamma)^2} \mathbb{E}_{s \sim d^{\pi_{\mathrm{E}}}} [\sqrt{\mathrm{KL}(\pi_{\mathrm{E}}(\cdot \mid s), \pi_{\mathrm{IQL}}(\cdot \mid s))}] \\
&\leq \frac{\sqrt{2}R_{\max}}{(1-\gamma)^2} \sqrt{\mathbb{E}_{s \sim d^{\pi_{\mathrm{E}}}} [\mathrm{KL}(\pi_{\mathrm{E}}(\cdot \mid s), \pi_{\mathrm{IQL}}(\cdot \mid s))]} \\
&\leq \frac{\sqrt{2M}R_{\max}}{(1-\gamma)^2} \sqrt{\mathbb{E}_{s \sim \mu} [\mathrm{KL}(\pi_{\mathrm{E}}(\cdot \mid s), \pi_{\mathrm{IQL}}(\cdot \mid s))]} = \frac{\sqrt{2M}R_{\max}}{(1-\gamma)^2} \sqrt{\epsilon_1},
\end{aligned}
\tag{17}
$$

where the second inequality uses Pinsker's inequality, the third inequality is obtained by Jensen's inequality, the fourth inequality uses Assumption 2, and the final equality follows from the definition of $\epsilon_1$.

**Imitation Error 2.** By the performance difference lemma (Lemma 6) and the same derivation of (11), we have

$$
\begin{aligned}
V^{\tilde{\pi}_{\mathrm{E}}} - V^{\tilde{\pi}_{\mathrm{IQL}}} &= \frac{R_{\max}}{1-\gamma} \mathbb{E}_{s \sim d^{\tilde{\pi}_{\mathrm{E}}}} \mathbb{E}_{a \sim \tilde{\pi}_{\mathrm{E}}(\cdot|s)} [A^{\tilde{\pi}_{\mathrm{IQL}}}(s, a)] \\
&\leq \frac{R_{\max}}{(1-\gamma)^2} \mathbb{E}_{s \sim d^{\tilde{\pi}_{\mathrm{E}}}} [\|\tilde{\pi}_{\mathrm{E}}(\cdot \mid s) - \tilde{\pi}_{\mathrm{IQL}}(\cdot \mid s)\|_1].
\end{aligned}
$$

By the same derivation of (17), we have

$$
V^{\tilde{\pi}_{\mathrm{E}}} - V^{\tilde{\pi}_{\mathrm{IQL}}} \leq \frac{\sqrt{2M}R_{\max}}{(1-\gamma)^2} \sqrt{\mathbb{E}_{s \sim \mathcal{D}} [\mathrm{KL}(\tilde{\pi}_{\mathrm{E}}(\cdot \mid s), \tilde{\pi}_{\mathrm{IQL}}(\cdot \mid s))]} \leq \frac{\sqrt{2M}R_{\max}}{(1-\gamma)^2} \sqrt{\epsilon_2}.
\tag{18}
$$

**Putting Together.** Combining (16), (17), and (18), we obtain that

$$
V^{\pi_{\mathrm{IQL}}} - V^{\tilde{\pi}_{\mathrm{IQL}}} \leq \frac{\sqrt{2M}R_{\max}}{(1-\gamma)^2} [\sqrt{\epsilon_1} + \sqrt{\epsilon_2}] + \frac{2R_{\max}}{(1-\gamma)^2} \sqrt{\frac{M\zeta}{N}},
$$

which concludes the proof of Theorem 3. $\qquad \square$

**Remark 7.** *We would like to make a comparison with Zhang et al. (2022), a theory work about offline RL with corrupted data. They assume that $\epsilon N$ data points are corrupted and propose a least-squares value iteration (LSVI) type algorithm that achieves an $\mathcal{O}(\sqrt{\epsilon})$ optimality gap by ignoring other parameters. When applying our results to their setting, we have $\zeta \leq \epsilon N$, which implies that our corruption error term $\mathcal{O}(\sqrt{\zeta/N}) \leq \mathcal{O}(\sqrt{\epsilon})$. This matches the result in Zhang et al. (2022), which combines LSVI with a robust regression oracle. However, their result may not readily apply to our scenario, given that we permit complete data corruption ($\epsilon = 1$). Furthermore, from the empirical side, IQL does not require any robust regression oracle and outperforms LSVI-type algorithms, such EDAC and MSG.*

## C.2 PROOF OF LEMMA 5

*Proof of Lemma 5.* We first state Theorem 1 of Sun et al. (2020) as follows.

**Lemma 8** (Theorem 1 of Sun et al. (2020)). *Consider the following statistical model:*

$$y_i = \langle x_i, \beta^* \rangle + \varepsilon_i, \quad with \, \mathbb{E}\left( \varepsilon_i \mid x_i \right) = 0, \mathbb{E}(|\varepsilon_i|^{1+\nu}) < \infty, \quad \forall 1 \leq i \leq N.$$

*By solving the Huber regression problem with a proper parameter $\delta$*

$$\hat{\beta} = \arg\min_{\beta} \sum_{i=1}^{N} l_H^{\delta}(y_i - \langle x_i, \beta \rangle),$$

*the obtained $\hat{\beta}$ satisfies*

$$\|\hat{\beta} - \beta^*\|_2 \lesssim N^{-\min\{\nu/(1+\nu), 1/2\}}.$$

Back to our proof, for any $(s_i, a_i, s_i') \sim \mathcal{D}$, we take (i) $x_i = \mathbb{1}\{s_i, a_i\} \in \mathbb{R}^{|S \times A|}$ as the one-hot vector at $(s_i, a_i)$ and (ii) $y_i = r_i + \gamma V_\psi(s_i')$. Moreover, we treat $\{r(s, a + \gamma \mathbb{E}_{s' \sim P(\cdot|s,a)}[V_\psi(s')]\}_{(s,a) \in S \times A} \in \mathbb{R}^{|S \times A|}$ as $\beta^*$. Hence, by Lemma 8 and the fact that $\|\cdot\|_\infty \leq \|\cdot\|_2$, we obtain

$$\|\hat{Q}(s, a) - r(s, a) - \gamma \mathbb{E}_{s' \sim P(\cdot|s,a)}[V_\psi(s')]\|_\infty \lesssim N^{-\min\{\nu/(1+\nu), 1/2\}},$$

where $N$ is the size of dataset $\mathcal{D}$. This concludes the proof of Lemma 5. $\qquad\square$

**A Discussion about Recovering the Optimal Value Function**  We denote the optimal solutions to (2) and (7) by $V_\tau$ and $Q_\tau$, respectively. When $N \to \infty$, we know

$$V_\tau(s) = \mathbb{E}_{a \sim \pi_{\mathcal{D}}(\cdot|s)}^{\tau}\left[Q_\tau(s, a)\right], \quad Q_\tau(s, a) = r(s, a) + \gamma \mathbb{E}_{s' \sim P(\cdot|s,a)}\left[V_\tau(s')\right],$$

where $\mathbb{E}_{x \sim X}^{\tau}[x]$ denotes the $\tau$-th expectile of the random variable $X$. following the analysis of IQL (Kostrikov et al., 2021, Theorem 3), we can further show that $\lim_{\tau \to 1} V_\tau(s) = \max_{a \in \mathcal{A} \text{ s.t. } \pi_{\mathcal{D}}(a|s) > 0} Q^*(s, a)$. With the Huber loss, we can further recover the optimal value function even in the presence of a heavy-tailed target distribution.

## D IMPLEMENTATION DETAILS

### D.1 DATA CORRUPTION DETAILS

We apply both random and adversarial corruption to the four elements, namely states, actions, rewards, and dynamics (or "next-states"). In our experiments, we primarily utilize the "medium-replay-v2" and "medium-expert-v2" datasets from (Fu et al., 2020). These datasets are gathered either during the training of a SAC agent or by combining equal proportions of expert demonstrations and medium data, making them more representative of real-world scenarios. To control the cumulative corruption level, we introduce two parameters, $c$ and $\epsilon$. Here, $c$ represents the corruption rate within the dataset of size $N$, while $\epsilon$ denotes the corruption scale for each dimension. We detail four types of random data corruption and a mixed corruption below:

- **Random observation attack**: We randomly sample $c \cdot N$ transitions $(s, a, r, s')$, and modify the state to $\hat{s} = s + \lambda \cdot \text{std}(s), \lambda \sim \text{Uniform}[-\epsilon, \epsilon]^{d_s}$. Here, $d_s$ represents the dimension of states and "std(s)" is the $d_s$-dimensional standard deviation of all states in the offline dataset. The noise is scaled according to the standard deviation of each dimension and is independently added to each respective dimension.

- **Random action attack**: We randomly select $c \cdot N$ transitions $(s, a, r, s')$, and modify the action to $\hat{a} = a + \lambda \cdot \text{std}(a), \lambda \sim \text{Uniform}[-\epsilon, \epsilon]^{d_a}$, where $d_a$ represents the dimension of actions and "std(a)" is the $d_a$-dimensional standard deviation of all actions in the offline dataset.

- **Random reward attack**: We randomly sample $c \cdot N$ transitions $(s, a, r, s')$ from $D$, and modify the reward to $\hat{r} \sim \text{Uniform}[-30 \cdot \epsilon, 30 \cdot \epsilon]$. We multiply by 30 because we have noticed that offline RL algorithms tend to be resilient to small-scale random reward corruption (as observed in (Li et al., 2023)), but would fail when faced with large-scale random reward corruption.

- **Random dynamics attack**: We randomly sample $c \cdot N$ transitions $(s, a, r, s')$, and modify the next-step state $\hat{s}' = s' + \lambda \cdot \text{std}(s'), \lambda \sim \text{Uniform}[-\epsilon, \epsilon]^{d_s}$. Here, $d_s$ indicates the dimension of states and "std$(s')$" is the $d_s$-dimensional standard deviation of all next-states in the offline dataset.

- **Random mixed attack**: We randomly select $c \cdot N$ of the transitions and execute the random observation attack. Subsequently, we again randomly sample $c \cdot N$ of the transitions and carry out the random action attack. The same process is repeated for both reward and dynamics attacks.

In addition, four types of adversarial data corruption are detailed as follows:

- **Adversarial observation attack**: We first pretrain an EDAC agent with a set of $Q_p$ functions and a policy function $\pi_p$ using clean dataset. Then, we randomly sample $c \cdot N$ transitions $(s, a, r, s')$, and modify the states to $\hat{s} = \min_{\hat{s} \in \mathbb{B}_d(s,\epsilon)} Q_p(\hat{s}, a)$. Here, $\mathbb{B}_d(s, \epsilon) = \{\hat{s} || \hat{s} - s| \le \epsilon \cdot \text{std}(s)\}$ regularizes the maximum difference for each state dimension. The Q function in the objective is the average of the Q functions in EDAC. The optimization is implemented through Projected Gradient Descent similar to prior works (Madry et al., 2017; Zhang et al., 2020). Specifically, We first initialize a learnable vector $z \in [-\epsilon, \epsilon]^{d_s}$, and then conduct a 100-step gradient descent with a step size of 0.01 for $\hat{s} = s + z \cdot \text{std}(s)$, and clip each dimension of $z$ within the range $[-\epsilon, \epsilon]$ after each update.

- **Adversarial action attack**: We use the pretrained EDAC agent with a group $Q_p$ functions and a policy function $\pi_p$. Then, we randomly sample $c \cdot N$ transitions $(s, a, r, s')$, and modify the actions to $\hat{a} = \min_{\hat{a} \in \mathbb{B}_d(a,\epsilon)} Q_p(s, \hat{a})$. Here, $\mathbb{B}_d(a, \epsilon) = \{\hat{a} || \hat{a} - a| \le \epsilon \cdot \text{std}(a)\}$ regularizes the maximum difference for each action dimension. The optimization is implemented through Projected Gradient Descent, as discussed above.

- **Adversarial reward attack**: We randomly sample $c \cdot N$ transitions $(s, a, r, s')$, and directly modify the rewards to: $\hat{r} = -\epsilon \times r$.

- **Adversarial dynamics attack**: We use the pretrained EDAC agent with a group of $Q_p$ functions and a policy function $\pi_p$. Then, we randomly select $c \cdot N$ transitions $(s, a, r, s')$, and modify the next-step states to $\hat{s}' = \min_{\hat{s}' \in \mathbb{B}_d(s',\epsilon)} Q_p(\hat{s}', \pi_p(\hat{s}'))$. Here, $\mathbb{B}_d(s', \epsilon) = \{\hat{s}' || \hat{s}' - s'| \le \epsilon \cdot \text{std}(s')\}$. The optimization is the same as discussed above.

Table 3: Hyperparameters used for RIQL under the random corruption benchmark.

| Environments | Attack Element | $N$ | $\alpha$ | $\delta$ |
|---|---|---|---|---|
| Halfcheetah | observation | 5 | 0.1 | 0.1 |
| | action | 3 | 0.25 | 0.5 |
| | reward | 5 | 0.25 | 3.0 |
| | dynamics | 5 | 0.25 | 3.0 |
| Walker2d | observation | 5 | 0.25 | 0.1 |
| | action | 5 | 0.1 | 0.5 |
| | reward | 5 | 0.1 | 3.0 |
| | dynamics | 3 | 0.25 | 1.0 |
| Hopper | observation | 3 | 0.25 | 0.1 |
| | action | 5 | 0.25 | 0.1 |
| | reward | 3 | 0.25 | 1.0 |
| | dynamics | 5 | 0.5 | 1.0 |

## D.2 IMPLEMENTATION DETAILS OF IQL AND RIQL

For the policy and value networks of IQL and RIQL, we utilize an MLP with 2 hidden layers, each consisting of 256 units, and ReLU activations. These neural networks are updated using the Adam

Table 4: Hyperparameters used for RIQL under the adversarial corruption benchmark.

| Environments | Attack Element | $N$ | $\alpha$ | $\delta$ |
|---|---|---|---|---|
| Halfcheetah | observation | 5 | 0.1 | 0.1 |
| | action | 5 | 0.1 | 1.0 |
| | reward | 5 | 0.1 | 1.0 |
| | dynamics | 5 | 0.1 | 1.0 |
| Walker2d | observation | 5 | 0.25 | 1.0 |
| | action | 5 | 0.1 | 1.0 |
| | reward | 5 | 0.1 | 3.0 |
| | dynamics | 5 | 0.25 | 1.0 |
| Hopper | observation | 5 | 0.25 | 1.0 |
| | action | 5 | 0.25 | 1.0 |
| | reward | 5 | 0.25 | 0.1 |
| | dynamics | 5 | 0.5 | 1.0 |

optimizer with a learning rate of $3 \times 10^{-4}$. We set the discount factor as $\gamma = 0.99$, the target networks are updated with a smoothing factor of 0.005 for soft updates. The hyperparameter $\beta$ and $\tau$ for IQL and RIQL are set to 3.0 and 0.7 across all experiments. In terms of policy parameterization, we argue that RIQL is robust to different policy parameterizations, such as deterministic policy and diagonal Gaussian policy. For the main results, IQL and RIQL employ a deterministic policy, which means that maximizing the weighted log-likelihood is equivalent to minimizing a weighted $l_2$ loss on the policy output: $\mathcal{L}_\pi(\phi) = \mathbb{E}_{(s,a)\sim\mathcal{D}}[\exp(\beta A(s,a))\|a - \pi_\phi(s)\|_2^2]$. We also include a discussion about the diagonal Gaussian parameterization in Appendix E.4. In the training phase, we train IQL, RIQL, and other baselines for $3 \times 10^6$ steps following (An et al., 2021), which corresponds to 3000 epochs with 1000 steps per epoch. The training is performed using a batch size of 256. For evaluation, we rollout each agent in the clean environment for 10 trajectories (maximum length equals 1000) and average the returns. All reported results are averaged over four random seeds. As for the specific hyperparameters of RIQL, we search $K \in \{3, \mathbf{5}\}$ and quantile $\alpha \in \{\mathbf{0.1}, \mathbf{0.25}, 0.5\}$ for the quantile Q estimator, and $\delta \in \{\mathbf{0.1}, 0.5, \mathbf{1.0}, 3.0\}$ for the Huber loss. In most settings, we find that the highlighted hyperparameters often yield the best results. The specific hyperparameters used for the random and adversarial corruption experiment in Section 6.1 are listed in Table 3 and Table 4, respectively. Our code is available at https://github.com/YangRui2015/RIQL, which is based on the open-source library of CORL (Tarasov et al., 2022).

### D.3    QUANTILE CALCULATION

To calculate the $\alpha$-quantile for a group of Q function $\{Q_{\theta_i}\}_{i=1}^K$, we can map $\alpha \in [0, 1]$ to the range of indices $[1, K]$ in order to determine the location of the quantile in the sorted input. If the quantile lies between two data points $Q_{\theta_i} < Q_{\theta_j}$ with indices $i, j$ in the sorted order, the result is computed according to the linear interpolation: $Q_\alpha = Q_{\theta_i} + (Q_{\theta_j} - Q_{\theta_i}) * \text{fraction}(\alpha \times (K-1) + 1)$, where the "fraction" represents the fractional part of the computed index. As a special case, when $K = 2$, $Q_\alpha = (1 - \alpha)\min(Q_{\theta_0}, Q_{\theta_1}) + \alpha\max(Q_{\theta_0}, Q_{\theta_1})$, and $Q_\alpha$ recovers the Clipped Double Q-learning trick when $\alpha = 0$.

## E    ADDITIONAL EXPERIMENTS

### E.1    ABLATION OF IQL

As observed in Figure 1, IQL demonstrates notable robustness against some types of data corruption. However, it raises the question: **which component of IQL contributes most to its robustness?** IQL can be interpreted as a combination of expectile regression and weighted imitation learning. To understand the contribution of each component, we perform an ablation study on IQL under mixed corruption in the Hopper and Walker tasks, setting the corruption rate to 0.1 and the corruption scale to 1.0. Specifically, we consider the following variants:

**IQL $\tau = \mathbf{0.7}$**: This is the standard IQL baseline.

**IQL $\tau = \mathbf{0.5}$**: This variant sets $\tau = 0.5$ for IQL.

**IQL w/o Expectile**: This variant removes the expectile regression that learns the value function and instead directly learns the Q function to minimize $\mathbb{E}_{(s,a,r,s')\sim\mathcal{D}}[(r+\gamma Q(s',\pi(s'))-Q(s,a))^2]$, and updating the policy to maximize $\mathbb{E}_{(s,a)\sim\mathcal{D}}[\exp(Q(s,a)-Q(s,\pi(s)))\log\pi(a|s)]$.

**ER w. Q gradient**: This variant retains the expectile regression to learn the Q function and V function, then uses only the learned Q function to perform a deterministic policy gradient: $\max_\pi[Q(s,\pi(s))]$.

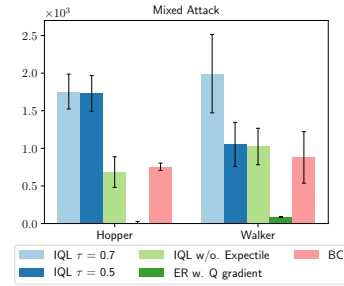

Figure 8: Ablation results of IQL under mixed attack.

The results are presented in Figure 8. From these, we can conclude that while expectile regression enhances performance, it is not the key factor for robustness. On the one hand, **IQL $\tau$ = 0.7** outperforms **IQL $\tau$ = 0.5** and **IQL w/o Expectile**, confirming that expectile regression is indeed an enhancement factor. On the other hand, the performance of **ER w. Q gradient** drops to nearly zero, significantly lower than BC, indicating that expectile regression is not the crucial component for robustness. Instead, the supervised policy learning scheme is proved to be the key to achieving better robustness.

### E.2    ABLATION OF OBSERVATION NORMALIZATION

Figure 9 illustrates the comparison between IQL and "IQL (norm)" on the medium-replay and medium-expert datasets under various data corruption scenarios. In most settings, IQL with normalization surpasses the performance of the standard IQL. This finding contrasts with the general offline Reinforcement Learning (RL) setting, where normalization does not significantly enhance performance, as noted by (Fujimoto & Gu, 2021). These results further justify our decision to incorporate observation normalization into RIQL.

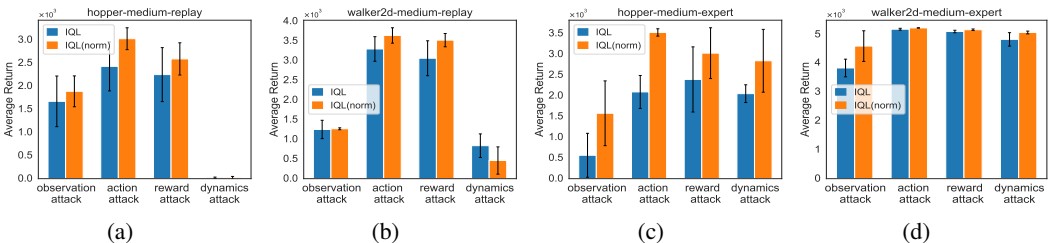

Figure 9: Comparison of IQL w/ and w/o observation normalization.

### E.3    EVALUATION UNDER THE MIXED CORRUPTION

We conducted experiments under mixed corruption settings, where corruption independently occurred on four elements: state, actions, rewards, and next-states (or dynamics). The results for corruption rates of 0.1/0.2 and corruption scales of 1.0/2.0 on the "medium-replay" datasets are presented in Figure 10 and Figure 11. From the figures, it is evident that **RIQL consistently outperforms other baselines by a significant margin in the mixed attack settings with varying corruption rates and scales**. Among the baselines, EDAC and MSG continue to struggle in this corruption setting. Besides, CQL also serves as a reasonable baseline, nearly matching the performance of IQL in such mixed corruption settings. Additionally, SQL also works reasonably under a small corruption rate of 0.1 but fails under a corruption rate of 0.2.

### E.4    EVALUATION OF DIFFERENT POLICY PARAMETERIZATION

In the official implementation of IQL (Kostrikov et al., 2021), the policy is parameterized as a diagonal Gaussian distribution with a state-independent standard deviation. However, in the data corruption setting, our findings indicate that this version of IQL generally underperforms IQL with a deterministic policy. This observation is supported by Figure 12, where we compare the performance under the mixed attack with a corruption rate of 0.1. Consequently, we use the deterministic

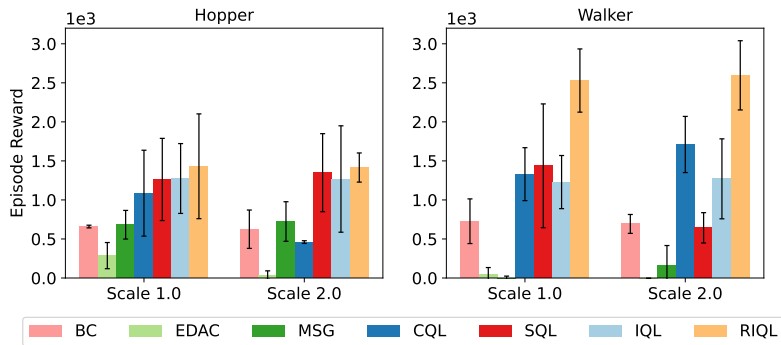

Figure 10: Results under random mixed attack with a corruption rate of 0.1.

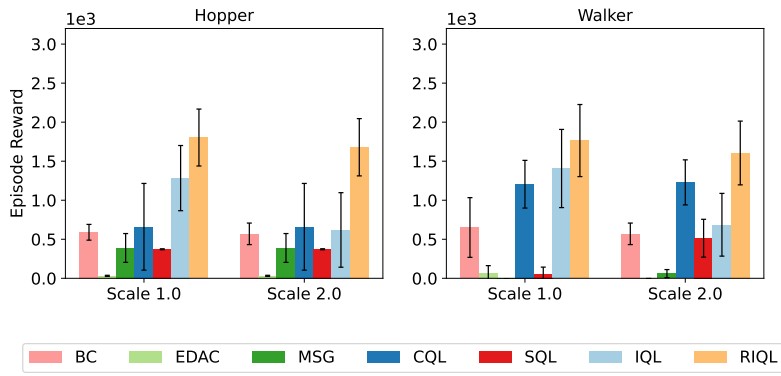

Figure 11: Results under random mixed attack with a corruption rate of 0.2.

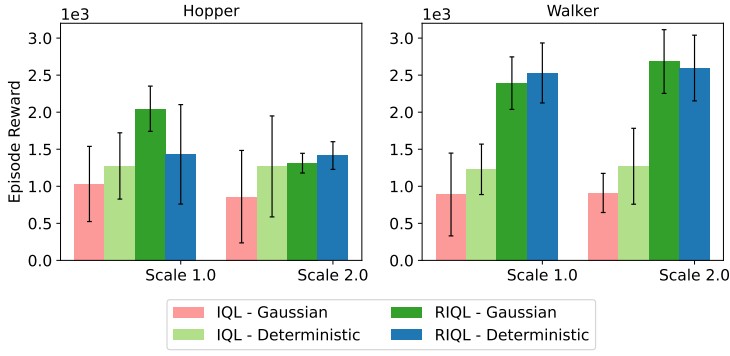

Figure 12: Comparison of different policy parameterizations under random mixed attack with a corruption rate of 0.1.

policy for both IQL and RIQL by default. Intriguingly, **RIQL demonstrates greater robustness to the policy parameterization**. In Figure 12, "RIQL - Gaussian" can even surpass "RIQL - Deterministic" and exhibits lower performance variance. We also note the consistent improvement of RIQL over IQL under different policy parameterizations, which serves as an additional advantage of RIQL. Based on our extensive experiments, we have noticed that "RIQL - Gaussian" offers greater stability and necessitates less hyperparameter tuning than "RIQL - Deterministic". Conversely, "RIQL - Deterministic" requires more careful hyperparameter tuning but can yield better results on average in scenarios with higher corruption rates.

Table 5: Average normalized performance under random data corruption (scale 1.0) using **"medium-expert" datasets**. Results are averaged over 4 random seeds.

| Environment | Attack Element | BC | EDAC | MSG | CQL | SQL | IQL | RIQL (ours) |
|---|---|---|---|---|---|---|---|---|
| Halfcheetah | observation | 51.6±2.9 | -4.0±3.2 | -4.9±4.5 | 3.4±1.6 | 30.5±5.1 | 71.0±3.4 | **79.7±7.2** |
| | action | 61.2±5.6 | 45.4±3.2 | 32.8±4.1 | 67.5±9.7 | 89.8±0.4 | 88.6±1.2 | **92.9±0.6** |
| | reward | 59.9±3.3 | 51.8±6.6 | 28.7±6.0 | 87.3±5.1 | 92.9±0.7 | **93.4±1.2** | 93.2±0.5 |
| | dynamics | 59.9±3.3 | 25.4±11.3 | 2.5±2.4 | 54.0±1.9 | 78.3±3.2 | 80.4±2.0 | **92.2±0.9** |
| Walker2d | observation | **106.2±1.7** | -0.4±0.0 | -0.3±0.1 | 15.0±2.6 | 17.5±4.3 | 71.1±23.5 | 101.3±11.8 |
| | action | 89.5±17.3 | 48.4±20.5 | 3.8±1.7 | 109.2±0.3 | 109.7±0.7 | 112.7±0.5 | **113.5±0.5** |
| | reward | 97.9±15.4 | 35.0±29.8 | -0.4±0.3 | 104.2±5.4 | 110.1±0.4 | 110.7±1.3 | **112.7±0.6** |
| | dynamics | 97.9±15.4 | 0.2±0.5 | 3.6±6.7 | 82.0±11.9 | 93.5±3.7 | 100.3±7.5 | **112.8±0.2** |
| Hopper | observation | 52.7±1.6 | 0.8±0.0 | 0.9±0.1 | 46.2±12.9 | 1.3±0.1 | 23.2±33.6 | **76.2±24.2** |
| | action | 50.4±1.5 | 23.5±6.0 | 18.8±10.6 | 83.5±13.5 | 59.1±32.0 | 77.8±16.4 | **92.6±38.9** |
| | reward | 52.4±1.5 | 0.7±0.0 | 3.9±5.4 | 90.8±3.6 | 85.4±16.3 | 78.6±31.0 | **91.4±25.9** |
| | dynamics | 52.4±1.5 | 5.6±6.5 | 2.1±2.4 | 26.7±7.5 | 75.8±9.3 | 69.7±25.3 | **84.6±15.4** |
| Average score ↑ | | 69.3 | 19.4 | 7.6 | 64.2 | 70.3 | 81.5 | **95.3** |

## E.5 Evaluation on the Medium-expert Datasets

In addition to the "medium-replay" dataset, which closely resembles real-world environments, we also present results using the "medium-expert" dataset under random corruption in Table 5. The "medium-expert" dataset is a mixture of $50\%$ expert and $50\%$ medium demonstrations. As a result of the inclusion of expert demonstrations, the overall performance of each algorithm is much better compared to the results obtained from the "medium-replay" datasets. Similar to the main results, EDAC and MSG exhibit very low average scores across various data corruption scenarios. Additionally, CQL and SQL only manage to match the performance of BC. IQL, on the other hand, outperforms BC by $17.6\%$, confirming its effectiveness. Most notably, **RIQL achieves the highest score in 10 out of 12 settings and improves by** $16.9\%$ **over IQL**. These results align with our findings in the main paper, further demonstrating the superior effectiveness of RIQL across different types of datasets.

Table 6: Average normalized performance under **random corruption of scale 2.0** using the "medium-replay" datasets. Results are averaged over 4 random seeds.

| Environment | Attack Element | BC | EDAC | MSG | CQL | SQL | IQL | RIQL (ours) |
|---|---|---|---|---|---|---|---|---|
| Walker2d | observation | 16.1±5.5 | -0.4±0.0 | -0.4±0.1 | 12.5±21.4 | 0.7±1.2 | 27.4±5.7 | **50.3±11.5** |
| | action | 13.3±2.5 | 77.5±3.4 | 16.3±3.7 | 15.4±9.1 | 79.0±5.2 | 69.9±2.8 | **82.9±3.8** |
| | reward | 16.0±7.4 | 1.2±1.6 | 9.3±5.9 | 48.4±4.6 | 0.2±0.9 | 56.0±7.4 | **81.5±4.6** |
| | dynamics | 16.0±7.4 | -0.1±0.0 | 4.2±3.8 | 0.1±0.4 | 18.5±3.6 | 12.0±5.6 | **77.8±6.3** |
| Hopper | observation | 17.6±1.2 | 2.7±0.8 | 7.1±5.8 | 29.2±1.9 | 13.7±3.1 | **69.7±7.0** | 55.8±17.4 |
| | action | 16.0±3.1 | 31.9±3.2 | 35.9±8.3 | 19.2±0.5 | 38.8±1.7 | 75.1±18.2 | **92.7±11.0** |
| | reward | 19.5±3.4 | 6.4±5.0 | 42.1±14.2 | 51.7±14.8 | 41.5±4.2 | 62.1±8.3 | **85.0±17.3** |
| | dynamics | 19.5±3.4 | 0.8±0.0 | 3.7±3.2 | 0.8±0.1 | 14.4±2.7 | 0.9±0.3 | **45.8±9.3** |
| Average score ↑ | | 16.8 | 15.0 | 14.8 | 22.2 | 25.9 | 46.6 | **71.6** |

Table 7: Average normalized performance under **adversarial corruption of scale 2.0** using the "medium-replay" datasets. Results are averaged over 4 random seeds.

| Environment | Attack Element | BC | EDAC | MSG | CQL | SQL | IQL | RIQL (ours) |
|---|---|---|---|---|---|---|---|---|
| Walker2d | observation | 13.0±1.5 | 3.9±6.7 | 3.5±3.9 | 63.3±5.1 | 2.7±3.0 | 47.0±8.3 | **71.5±6.4** |
| | action | 0.3±0.5 | **9.2±2.4** | 4.8±0.4 | 1.7±3.5 | 5.6±1.4 | 3.0±0.9 | 7.9±0.5 |
| | reward | 16.0±7.4 | -0.1±0.0 | 9.1±3.8 | 28.4±19.2 | -0.3±0.0 | 22.9±12.5 | **81.1±2.4** |
| | dynamics | 16.0±7.4 | 2.2±1.8 | 1.9±0.2 | 4.2±2.2 | 3.4±1.9 | 1.7±1.0 | **66.0±7.7** |
| Hopper | observation | 17.0±5.4 | 23.9±10.6 | 19.9±6.6 | 52.6±8.8 | 14.0±1.2 | 44.3±2.9 | **57.0±7.6** |
| | action | 9.9±4.6 | 23.3±4.7 | 22.8±1.7 | 14.7±0.9 | 19.3±3.1 | 21.8±4.9 | **23.8±4.0** |
| | reward | 19.5±3.4 | 1.2±0.5 | 22.9±1.1 | 22.9±1.1 | 0.6±0.0 | 24.7±1.6 | **35.5±6.7** |
| | dynamics | 19.5±3.4 | 0.6±0.0 | 0.6±0.0 | 0.6±0.0 | 24.3±3.2 | 0.7±0.0 | **29.9±1.5** |
| Average score ↑ | | 13.9 | 8.0 | 10.7 | 23.6 | 8.7 | 20.8 | **46.6** |

### E.6 EVALUATION UNDER LARGE-SCALE CORRUPTION

In our main results, we focused on random and adversarial corruption with a scale of 1.0. In this subsection, we present an empirical evaluation under a corruption scale of 2.0 using the "medium-replay" datasets. The results for random and adversarial corruptions are presented in Table 6 and Table 7, respectively.

In both tables, **RIQL achieves the highest performance in 7 out of 8 settings, surpassing IQL by** $53.5\%$ **and** $124.0\%$, respectively. Most algorithms experience a significant decrease in performance under adversarial corruption at the same scale, with the exception of CQL. However, it is still not comparable to our algorithm, RIQL. These results highlight the superiority of RIQL in handling large-scale random and adversarial corruptions.

### E.7 TRAINING TIME

We report the average epoch time on the Hopper task as a measure of computational cost in Table 8. From the table, it is evident that BC requires the least training time, as it has the simplest algorithm design. IQL requires more than twice the amount of time compared to BC, as it incorporates expectile regression and weighted imitation learning. Additionally, RIQL requires a comparable computational cost to IQL, introducing the Huber loss and quantile Q estimators. On the other hand, EDAC, MSG, and CQL require significantly longer training time. This is primarily due to their reliance on a larger number of Q ensembles and additional computationally intensive processes, such as the approximate logsumexp via sampling in CQL. Notably, DT necessitates the longest epoch time, due to its extensive transformer-based architecture. The results indicate that RIQL achieves significant gains in robustness without imposing heavy computational costs.

Table 8: Average Epoch Time.

| Algorithm | BC | DT | EDAC | MSG | CQL | IQL | RIQL |
|---|---|---|---|---|---|---|---|
| Time (s) | 3.8±0.1 | 28.9±0.2 | 14.1±0.3 | 12.0±0.8 | 22.8±0.7 | 8.7±0.3 | 9.2±0.4 |

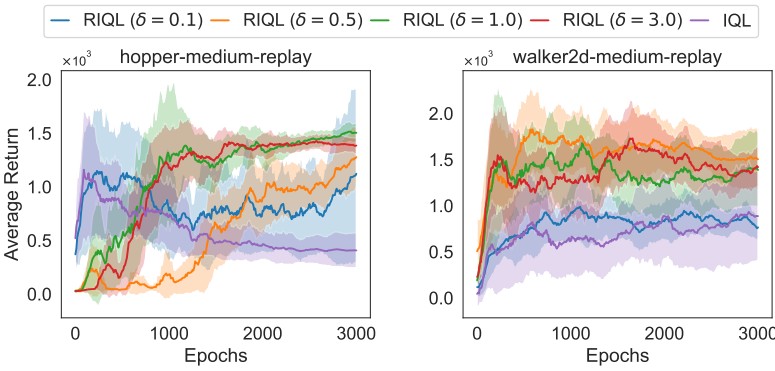

Figure 13: Hyperparameter study for $\delta$ in the Huber loss under the mixed attack.

### E.8 HYPERPARAMETER STUDY

In this subsection, we investigate the impact of key hyperparameters in RIQL, namely $\delta$ in the Huber loss, and $\alpha$, $K$ in the quantile Q estimators. We conduct our experiments using the mixed attack, where corruption is independently applied to each element (states, actions, rewards, and next-states) with a corruption rate of $0.2$ and a corruption scale of $1.0$. The dataset used for this evaluation is the "medium-replay" dataset of the Hopper and Walker environments.

**Hyperparameter** $\delta$    In this evaluation, we set $\alpha = 0.1$ and $K = 5$ for RIQL. The value of $\delta$ directly affects the position of the boundary between the $l_1$ loss and the $l_2$ loss. As $\delta$ approaches

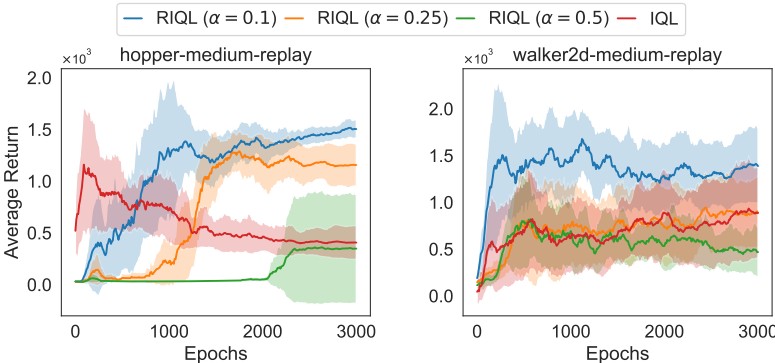

Figure 14: Hyperparameter study for $\alpha$ in the quantile estimators under the mixed attack.

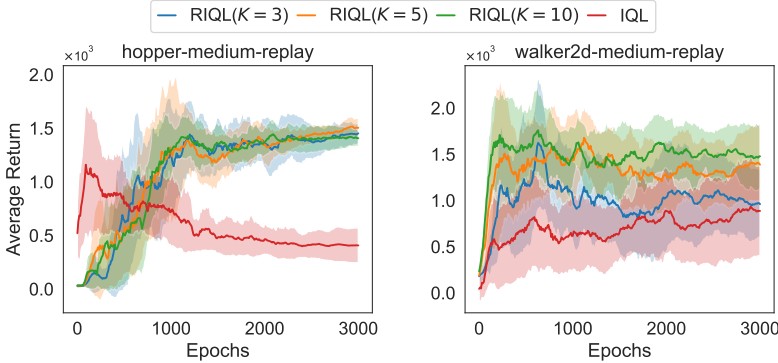

Figure 15: Hyperparameter study for $K$ in the quantile estimators under the mixed attack.

0, the boundary moves far from the origin of coordinates, making the loss function similar to the squared loss and resulting in performance closer to IQL. This can be observed in Figure 13, where $\delta = 0.1$ is the closest to IQL. As $\delta$ increases, the Huber loss becomes more similar to the $l_1$ loss and exhibits greater robustness to heavy-tail distributions. However, excessively large values of $\delta$ can decrease performance, as the $l_1$ loss can also impact the convergence. This is evident in the results, where $\delta = 3.0$ slightly decreases performance. Generally, $\delta = 1.0$ achieves the best or nearly the best performance in the mixed corruption setting. However, it is important to note that the optimal value of $\delta$ also depends on the corruption type when only one type of corruption (e.g., state, action, reward, and next-state) takes place. For instance, we find that $\delta = 0.1$ generally yields the best results for state and action corruption, while $\delta = 1$ is generally optimal for the dynamics corruption.

**Hyperparameter $\alpha$** In this evaluation, we set $\delta = 1.0$ and $K = 5$ for RIQL. The value of $\alpha$ determines the quantile used for estimating Q values and advantage values. A small $\alpha$ results in a higher penalty for the corrupted data, generally leading to better performance. This is evident in Figure 14, where $\alpha = 0.1$ achieves the best performance in the mixed corruption setting. However, as mentioned in Section 5.3, when only dynamics corruption is present, it is advisable to use a slightly larger $\alpha$, such as $0.25$, to prevent the excessive pessimism in the face of dynamics corruption.

**Hyperparameter $K$** In this evaluation, we set $\alpha = 0.1$ and $\delta = 1.0$ for RIQL. Hyperparameter $K$ is used to adjust the quantile Q estimator for in-dataset penalty. From Figure 15, we can observe that the impact of $K$ is more pronounced in the walker2d task compared to the hopper task, whereas the above hyperparameter $\alpha$ has a greater influence on the hopper task. For the walker2d task, a larger value of $K$ leads to better performance. Overall, $K = 5$ achieves the best or nearly the best performance in both settings. Therefore, we set $K = 5$ by default, considering both computational cost and performance.

Table 9: Comparison with **additional baselines under random data corruption**. Average normalized scores are reported and the highest score is highlighted.

| Environment | Attack Element | DT | MSG | UWMSG | RIQL (ours) |
|---|---|---|---|---|---|
| Halfcheetah | observation | **33.7±1.9** | -0.2±2.2 | 2.1±0.4 | 27.3±2.4 |
| | action | 36.3±1.5 | 52.0±0.9 | **53.8±0.6** | 42.9±0.6 |
| | reward | 39.2±1.0 | 17.5±16.4 | 39.9±1.8 | **43.6±0.6** |
| | dynamics | 33.7±1.9 | 1.7±0.4 | 3.6±2.0 | **43.1±0.2** |
| Walker2d | observation | **54.5±5.0** | -0.4±0.1 | 1.5±2.0 | 28.4±7.7 |
| | action | 10.3±3.1 | 25.3±10.6 | 61.2±9.9 | **84.6±3.3** |
| | reward | 65.5±4.9 | 18.4±9.5 | 65.9±13.9 | **83.2±2.6** |
| | dynamics | 54.5±5.0 | 7.4±3.7 | 6.5±2.8 | **78.2±1.8** |
| Hopper | observation | **65.2±12.1** | 6.9±5.0 | 11.3±4.0 | 62.4±1.8 |
| | action | 15.5±0.7 | 37.6±6.5 | 82.0±17.6 | **90.6±5.6** |
| | reward | 78.0±5.3 | 24.9±4.3 | 61.0±19.5 | **84.8±13.1** |
| | dynamics | **65.2±12.1** | 12.4±4.9 | 18.7±4.6 | 51.5±8.1 |
| Average score ↑ | | 46.0 | 17.0 | 34.0 | **60.0** |

## E.9 ADDITIONAL BASELINES

In addition to the baselines compared in the main paper, we compare RIQL to a sequence-modeling baseline, Decision Transformer (DT) (Chen et al., 2021), and a recently proposed robust offline RL algorithm for data corruption, UWMSG (Ye et al., 2023b). The results are presented in Table 9 and Table 10. Overall, RIQL demonstrates robust performance compared to the two additional baselines.

In comparison to BC, DT demonstrates a notable improvement, highlighting the effectiveness of sequence modeling. Furthermore, as DT employs a distinct sequence modeling approach from TD-based methods, it does not store data in independent transitions $\{s_i, a_i, r_i, s'_i\}_{i=1}^N$. Instead, it utilizes a trajectory sequence $\{s_t^i, a_t^i, r_t^i\}_{t=1}^T$ for $i$-th trajectory. Consequently, the observation corruption and dynamics corruption are identical for DT. Analyzing Table 9 and Table 10, we can observe that DT holds an advantage under observation corruption due to its ability to condition on historical information rather than relying solely on a single state. Despite leveraging trajectory history information and requiring 3.1 times the epoch time compared to RIQL, DT still falls significantly behind RIQL in performance under both random and adversarial data corruption. Exploring the incorporation of trajectory history, similar to DT, into RIQL could be a promising avenue for future research.

With regard to UWMSG, we observe its improvement over MSG, confirming its effectiveness with uncertainty weighting. However, it is still more vulnerable to data corruption compared to RIQL. We speculate that the explicit uncertainty estimation used in UWMSG is still unstable.

Table 10: Comparison with **additional baselines under adversarial data corruption**. Average normalized scores are reported and the highest score is highlighted.

| Environment | Attack Element | DT | MSG | UWMSG | RIQL (ours) |
|---|---|---|---|---|---|
| Halfcheetah | observation | 35.5±4.3 | 1.1±0.2 | 1.3±0.8 | **35.7±4.2** |
| | action | 16.6±1.2 | 37.3±0.7 | **39.4±3.1** | 31.7±1.7 |
| | reward | 37.0±2.6 | **47.7±0.4** | 43.8±0.8 | 44.1±0.8 |
| | dynamics | 35.5±4.3 | -1.5±0.0 | 10.9±1.8 | **35.8±2.1** |
| Walker2d | observation | 57.4±3.2 | 2.9±2.7 | 9.9±1.5 | **70.0±5.3** |
| | action | 9.5±1.6 | 5.4±0.9 | 7.4±2.4 | **66.1±4.6** |
| | reward | 64.9±5.4 | 9.6±4.9 | 52.9±12.1 | **85.0±1.5** |
| | dynamics | 57.4±3.2 | 0.1±0.2 | 2.1±0.2 | **60.6±21.8** |
| Hopper | observation | **57.7±15.1** | 16.0±2.8 | 13.6±0.7 | 50.8±7.6 |
| | action | 14.0±1.1 | 23.0±2.1 | 34.0±4.4 | **63.6±7.3** |
| | reward | **68.6±10.5** | 22.6±2.8 | 23.6±2.1 | 65.8±9.8 |
| | dynamics | 57.7±15.1 | 0.6±0.0 | 0.8±0.0 | **65.7±21.1** |
| Average score ↑ | | 42.7 | 13.7 | 20.0 | **56.2** |

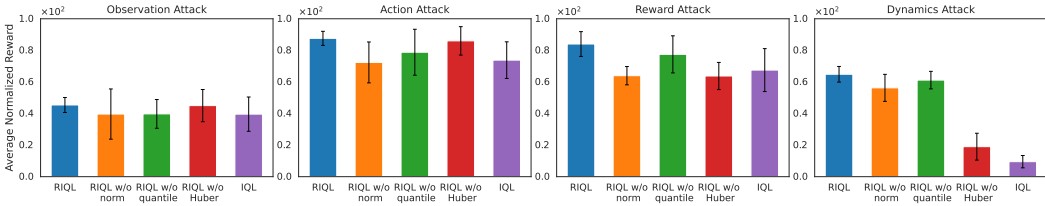

Figure 16: Ablations of RIQL under four types of data corruption.

### E.10 ADDITIONAL ABLATION STUDY

In this subsection, we analyze the contributions of RILQ's components in the presence of observation, action, reward, and dynamics corruption. Figure 16 illustrates the average normalized performance of Walker and Hopper tasks on the medium-replay datasets. We consider three variants of RIQL: RIQL without observation normalization (**RIQL w/o norm**), RIQL without the quantile Q estimator (**RIQL w/o quantile**), and RIQL without the Huber loss (**RIQL w/o Huber**). In **RIQL w/o quantile**, the Clipped Double Q trick used in IQL, is employed to replace the quantile Q estimator in RIQL. Based on the figure, we draw the following conclusions for each component: (1) normalization is beneficial for all four types of corruption, (2) quantile Q estimator is more effective for observation and action corruption but less so for reward and dynamics corruption, (3) conversely, the Huber loss is more useful for reward and dynamics corruption but less effective for observation and action corruption. Overall, all three components contribute to the performance of RIQL.

## F EVALUATION ON ANTMAZE TASKS

In this section, we present the evaluation of RIQL and baselines on a more challenging benchmark, AntMaze, which introduces additional difficulties due to its sparse-reward setting. Specifically, we assess four "medium" and "large" level tasks. Similar to the MuJoCo tasks, we consider random data corruption, but with a corruption rate of 0.2 for all AntMaze tasks. We observe that algorithms in AntMaze tasks are more sensitive to data noise, particularly when it comes to observation and dynamics corruption. Therefore, we use smaller corruption scales for observation (0.3), action (1.0), reward (30.0), and dynamics (0.3). The hyperparameters used for RIQL are list in Talbe 12.

As shown in Table 11, methods such as BC, DT, EDAC, and CQL struggle to learn meaningful policies in the presence of data corruption. In contrast, IQL exhibits relatively higher robustness, particularly under action corruption. RIQL demonstrates a significant improvement of 77.8% over IQL, successfully learning reasonable policies even in the presence of all types of corruption. These results further strengthen the findings in our paper and highlight the importance of RIQL.

Table 11: Average normalized performance under random data corruption on AntMaze tasks.

| Environment | Attack Element | BC | DT | EDAC | CQL | IQL | RIQL (ours) |
|---|---|---|---|---|---|---|---|
| antmaze-medium-play-v2 | observation | 0.0±0.0 | 0.0±0.0 | 0.0±0.0 | 0.8±1.4 | 7.4±5.0 | **18.0 ± 4.7** |
| | action | 0.0±0.0 | 0.0±0.0 | 0.0±0.0 | 0.8±1.4 | 63.5 ± 5.4 | **64.6 ± 5.2** |
| | reward | 0.0±0.0 | 0.0±0.0 | 0.0±0.0 | 0.0±0.0 | 12.4 ± 8.1 | **61.0 ± 5.8** |
| | dynamics | 0.0±0.0 | 0.0±0.0 | 0.0±0.0 | 37.5±28.6 | 32.7 ± 8.4 | **63.2 ± 5.4** |
| antmaze-medium-diverse-v2 | observation | 0.0±0.0 | 0.0±0.0 | 0.0±0.0 | 0.0±0.0 | 11.9 ± 6.9 | **12.5 ± 7.9** |
| | action | 0.0±0.0 | 0.0±0.0 | 0.0±0.0 | 15.0±22.3 | **65.6 ± 4.6** | 62.3 ± 5.5 |
| | reward | 0.0±0.0 | 0.0±0.0 | 0.0±0.0 | 0.0±0.0 | 8.3 ± 4.5 | **22.9 ± 17.9** |
| | dynamics | 0.0±0.0 | 0.0±0.0 | 0.0±0.0 | 20.8±14.2 | 35.5 ± 8.5 | **41.8 ± 7.2** |
| antmaze-large-play-v2 | observation | 0.0±0.0 | 0.0±0.0 | 0.0±0.0 | 0.0±0.0 | 0.0±0.0 | **27.3 ± 5.1** |
| | action | 0.0±0.0 | 0.0±0.0 | 0.0±0.0 | 0.0±0.0 | 33.1 ± 6.2 | **33.2 ± 6.3** |
| | reward | 0.0±0.0 | 0.0±0.0 | 0.0±0.0 | 0.0±0.0 | 0.6 ± 0.9 | **27.7 ± 8.0** |
| | dynamics | 0.0±0.0 | 0.0±0.0 | 0.0±0.0 | 4.2±5.5 | 1.6 ± 1.5 | **25.0 ± 6.4** |
| antmaze-large-diverse-v2 | observation | 0.0±0.0 | 0.0±0.0 | 0.0±0.0 | 0.0±0.0 | 0.0±0.0 | **24.9±10.7** |
| | action | 0.0±0.0 | 0.0±0.0 | 0.0±0.0 | 0.0±0.0 | **33.7 ± 5.9** | 30.9 ± 9.4 |
| | reward | 0.0±0.0 | 0.0±0.0 | 0.0±0.0 | 0.0±0.0 | 1.1 ± 1.1 | **20.2 ± 17.0** |
| | dynamics | 0.0±0.0 | 0.0±0.0 | 0.0±0.0 | 4.2±7.2 | 2.5 ± 2.9 | **16.5±5.7** |
| Average score ↑ | | 0.0 | 0.0 | 0.0 | 5.2 | 19.4 | **34.5** |

Table 12: Hyperparameters used for RIQL under the random corruption on the AntMaze Tasks.

| Environments | Attack Element | $N$ | $\alpha$ | $\delta$ |
|---|---|---|---|---|
| antmaze-medium-play-v2 | observation | 5 | 0.25 | 0.5 |
| | action | 3 | 0.5 | 0.1 |
| | reward | 3 | 0.5 | 0.5 |
| | dynamics | 3 | 0.5 | 0.5 |
| antmaze-medium-diverse-v2 | observation | 3 | 0.25 | 0.1 |
| | action | 3 | 0.5 | 0.1 |
| | reward | 3 | 0.25 | 0.1 |
| | dynamics | 3 | 0.5 | 0.1 |
| antmaze-large-play-v2 | observation | 3 | 0.25 | 0.5 |
| | action | 3 | 0.25 | 0.1 |
| | reward | 3 | 0.5 | 0.5 |
| | dynamics | 3 | 0.5 | 1.0 |
| antmaze-large-diverse-v2 | observation | 3 | 0.25 | 0.5 |
| | action | 5 | 0.25 | 0.05 |
| | reward | 3 | 0.25 | 0.5 |
| | dynamics | 3 | 0.5 | 0.5 |

### F.1 Learning Curves of the Benchmark Experiments

In Figure 17 and Figure 18, we present the learning curves of RIQL and other baselines on the "medium-replay" datasets, showcasing their performance against random and adversarial corruption, respectively. The corruption rate is set to 0.3 and the corruption scale is set to 1.0. These results correspond to Section 6.1 in the main paper. From these figures, it is evident that RIQL not only attains top-level performance but also exhibits remarkable learning stability compared to other baselines that can experience significant performance degradation during training. We postulate that this is attributed to the inherent characteristics of the Bellman backup, which can accumulate errors along the trajectory throughout the training process, particularly under the data corruption setting. On the contrary, **RIQL can effectively defend such effect and achieves both exceptional performance and remarkable stability**.

## G   Discussion of the Corruption Setting

In our corruption setting, the states and next-states are independently corrupted. However, our work does not consider the scenario where next-states are not explicitly saved in the dataset, meaning that corrupting a next-state would correspond to corrupting another state in the same trajectory.

The motivations behind our chosen corruption setting are as follows:

- Our work aims to address a comprehensive range of data corruption types. Consequently, conducting an independent analysis of each element's vulnerability can effectively highlight the disparities in their susceptibility.

- When data are saved into $(s, a, r, s')$ for offline RL, each element in the tuple can be subject to corruption independently.

- Our corruption method aligns with the concept of $\epsilon$-Contamination in offline RL theoretical analysis (Zhang et al., 2022), which considers arbitrary corruption on the 4-tuples $(s, a, r, s')$.

- Data corruption can potentially occur at any stage, such as storage, writing, reading, transmission, or processing of data. When data is loaded into memory, there is also a possibility of memory corruption and even intentional modifications by hackers. In the case of TD-based offline RL methods, it is common practice to load 4-tuples $(s, a, r, s')$ into memory and each element has the potential to be corrupted independently.

In the future, exploring the effects of data corruption in scenarios considering the dependence between states and next-states, or in the context of POMDP, holds promise for further research.

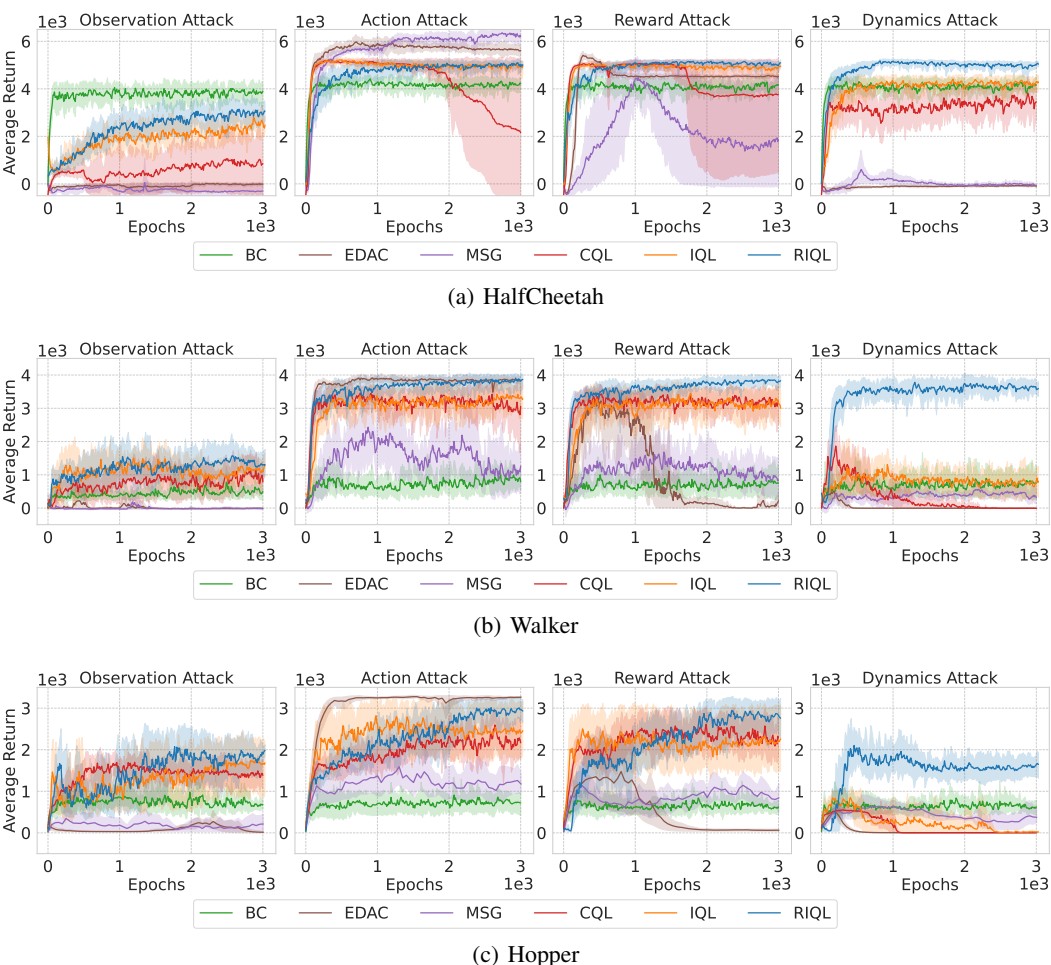

Figure 17: Learning curves under **random data corruption** on the "medium-replay" datasets.

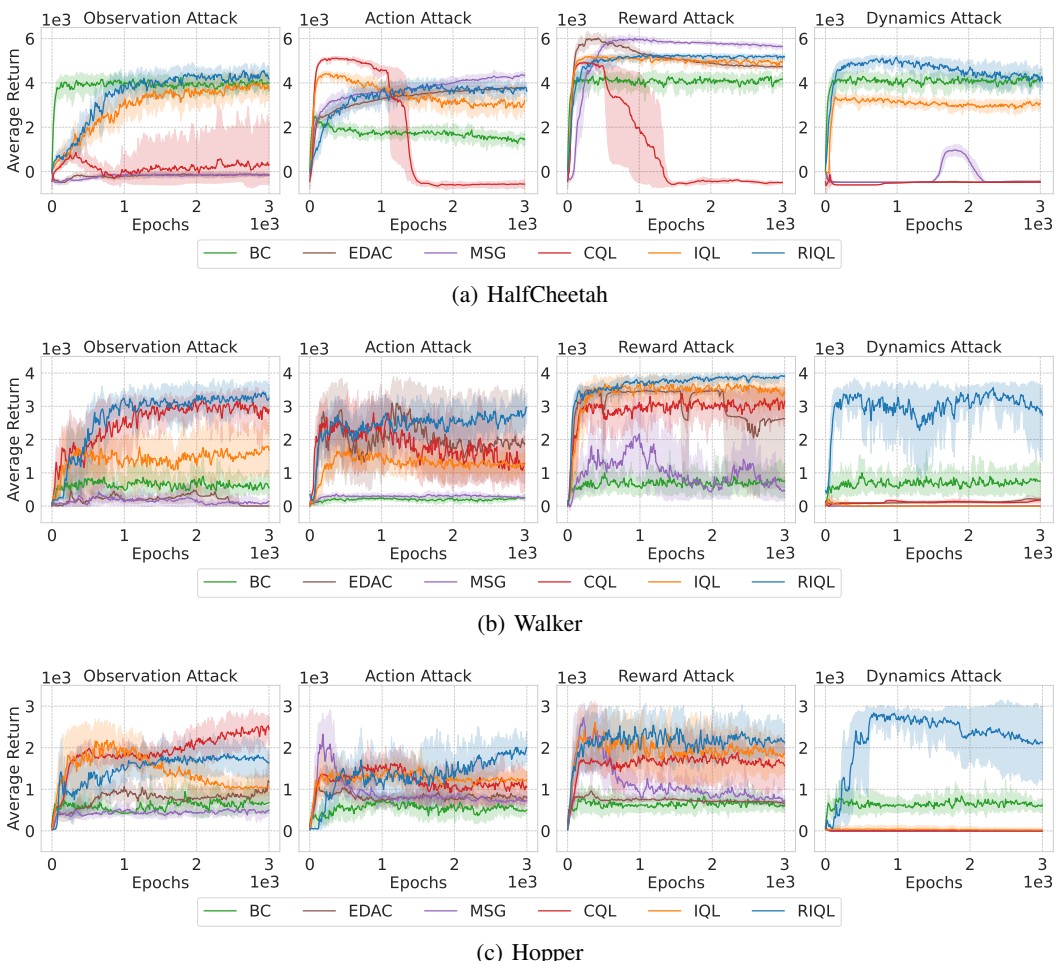

Figure 18: Learning curves under **adversarial data corruption** on the "medium-replay" datasets.

