# OpenReview forum: "Towards Robust Offline Reinforcement Learning under Diverse Data Corruption"
_ICLR.cc/2024/Conference — ICLR 2024 spotlight_

### Official Review · Reviewer_wKTe · 2023-10-27

**Soundness:** 3 good
**Presentation:** 3 good
**Contribution:** 3 good
**Rating:** 8
**Confidence:** 5

**Summary:**

In this paper, the authors investigate the robustness of offline RL algorithms when dealing with data corruption, including states, actions, rewards, and dynamics.
They then introduce Robust IQL (RIQL), an offline RL algorithm that enhances robustness through observation normalization, Huber loss, and quantile Q estimators.
Empirical evaluations demonstrate RIQL's exceptional resistance to various data corruption types, both random and adversarial.
This research sheds light on the vulnerability of current offline RL algorithms and provides a promising solution to enhance their robustness in real-world scenarios.

**Strengths:**

+ This study illuminates the susceptibility of existing offline RL algorithms while offering a promising approach to bolster their resilience in real-world environments. The proposed analysis is unquestionably relevant to the offline RL community. While there have been previous works (e.g., Zhang et al. (2022)) that offer theoretical analyses regarding the impact of using contaminated data in offline RL algorithms, this work appears to take it a step further by examining the impact of various forms of contamination (actions, states, etc.) and providing a highly detailed analysis for the offline IQL model. As a conclusion, it suggests a robust alternative for IQL, addressing how to mitigate its identified weaknesses.


+ The theoretical and experimental analysis presented in Section 3 is elegantly and clearly articulated. The authors' discovery about the robustness of using a weighted imitation learning strategy is intriguing, but for the specific case of dynamics corruption, this doesn't seem to hold true. This observation leads them to propose up to three modifications to the IQL model to enhance its robustness. Each of these modifications (observation normalization, Huber loss, and quantile Q estimators) is well-founded and justified.

+ The experimental evaluation is comprehensive and very detailed. First, all state-of-the-art models in offline-RL are assessed to see how they perform in perturbation scenarios. Once the analysis is completed, the article focuses on improvements for the IQL model. The environments used for the experiments are well-known within the community (Halfcheetah, Halfcheetah, Halfcheetah). The results presented are conclusive. I also appreciate the ablation study in section 5.3, which allows the reader to understand the impact of the modifications applied to IQL until achieving the robust version (RIQL).

**Weaknesses:**

- Some particularly interesting, and I would say more challenging, environments have been left out of the experimental analysis, especially in terms of the potential impact of perturbations, and in which the original IQL model was tested. It would be interesting to hear the authors' opinion on this. I'm referring to the following environments: locomotion-v2, antmaze-xx, kitchen, adroit.

- Somewhat, the theoretical analysis and the experimental evidence detailed in the section appear contradictory. The paper explicitly states this as follows: "Our theoretical analysis suggests that the adoption of weighted imitation learning inherently offers robustness under data corruption. However, empirical results indicate that IQL still remains susceptible to dynamics attacks".  This contradiction is not further examined in the manuscript. On the contrary, what is proposed is to address IQL's issue with dynamic attacks.

- Previous works on certification protocols for offline RL against attacks has not been considered for the novel RIQL model proposed. This is an important weaknesses that should be addressed in the rebuttal.

- Minor comments:
(Wu, 2022) Update the reference as it is not anymore an ArXiv manuscript but an ICLR22 paper.

**Questions:**

- As I pointed above, Have the authors evaluated how RIQL performs in any of the environments I mentioned above?
- Why hasn't the new model RIQL been tested on the COPA protocol in (Wu, 2022)?

---

> ### Author Response · Authors · 2023-11-17
> **Response to Reviewer wKTe**
>
> We thank the reviewer for the constructive comments, and we provide clarification to your concerns as follows. We would appreciate it if you have any further questions or comments.
>
> **Q1: more challenging, environments have been left out of the experimental analysis, especially in terms of the potential impact of perturbations ... I'm referring to the following environments: locomotion-v2, antmaze-xx, kitchen, adroit.**
>
> **A1:** Thank you for your suggestion. Our initial focus was to thoroughly study data corruption in commonly used MuJoCo environments. However, as the reviewer pointed out, we lacked experiments on more challenging tasks. **To address this concern, we conducted experiments on the AntMaze benchmark during the rebuttal period.** The results presented in Table 11 in Appendix I indicate that AntMaze poses a greater challenge compared to the MuJoCo benchmark. Most offline RL algorithms struggle to learn from corrupted datasets in these environments, primarily due to the high dimensionality of states and the issue of sparse rewards. In contrast, our proposed algorithm RIQL demonstrates superior effectiveness in learning policies from corrupted data, surpassing other algorithms significantly. These results further strengthen the findings in our paper and highlight the importance of RIQL. For a quick overview, we also provide the average results across four AntMaze environments below.
>
>
>
> | Attack Element |  BC  |  DT  | EDAC | CQL  |  IQL  | RIQL |
> |:--------------:|:----:|:----:|:----:|:----:|:----:|:-------:|
> |  Observation   |  0.0 |  0.0 |  0.0 |  0.2 |  4.8 | **20.7** |
> |     Action     |  0.0 |  0.0 |  0.0 |  4.0 | **49.0** |   47.8  |
> |     Reward     |  0.0 |  0.0 |  0.0 |  0.0 |  5.6 | **33.0** |
> |    Dynamics    |  0.0 |  0.0 |  0.0 | 16.7 | 18.1 | **36.6** |
> |     Average    |  0.0 |  0.0 |  0.0 |  5.2 | 19.4 | **34.5** |
>
>
> **Q2: the theoretical analysis and the experimental evidence detailed in the section appear contradictory ... empirical results indicate that IQL still remains susceptible to dynamics attacks. This contradiction is not further examined in the manuscript. On the contrary, what is proposed is to address IQL's issue with dynamic attacks.**
>
> **A2:**  Thank you for the valuable question. In fact, **our theoretical analysis not only aligns with our empirical findings but also provides an explanation for why IQL remains susceptible to dynamics corruption.** Specifically, the corruption error term in our theory scales as $\sqrt{\zeta/N}$ (ignoring other dependencies on $R_{\max}, \gamma$, and $M$), where $\zeta = \sum_{i=1}^N (\zeta_i + \log \zeta_i')$ is the cumulative corruption level. This suggests that IQL works well for data corruption scenarios when $\zeta \ll N$. **In the dynamics corruption scenario, the heavy-tailed issue can cause $\\{\zeta_i\\}\_{i=1}^N$ to become very large ($\\|\cdot \\|_{\infty}$ over a heavy-tailed distribution can be very large, see definition of $\zeta_i$ in Assumption 1), thereby making the upper bound in Theorem 3 very loose. This can explain IQL's susceptibility under dynamics corruption.**
> Furthermore, we would like to highlight that addressing the heavy-tailed issue via Huber regression considerably enhances IQL's performance. This, to a certain extent, confirms that the heavy-tailed issue is a crucial factor leading to IQL's poor performance, as the Huber loss serves as an effective method for tackling heavy-tailed regression problems (Lemma 5, Section 5.2).
> We have added this discussion in Section 5.2, as highlighted in blue.
>
>
> **Q3: Previous works on certification protocols for offline RL against attacks has not been considered for the novel RIQL model proposed. This is an important weaknesses that should be addressed in the rebuttal.**
>
> **A3:** Thank you for the suggestion. Upon thoroughly examining related works, we discovered that COPA [1] is the only study focusing on the certified robustness of offline RL with data corruption. However, **COPA is not directly applicable to our setting due to several differences.** Specifically, COPA corrupts trajectories, while we corrupt independent transitions. Additionally, COPA's certification protocol necessitates a discrete action space, whereas our investigated environments are continuous. As proposing a new certification protocol is beyond the scope of our paper, we have not conducted certification. Meanwhile, Theorem 3 in our paper appears to offer some form of "certification" for the policy learned by IQL since it guarantees that IQL achieves reasonable performance when the offline dataset is corrupted.
>
>
> **Q4: Minor comments: (Wu, 2022) Update the reference as it is not anymore an ArXiv manuscript but an ICLR22 paper.**
>
> **A4:** Thank you for bringing this mistake to our attention. We have revised this in the revision.
>
> **References**
>
> [1] Wu F, et al. COPA: Certifying Robust Policies for Offline Reinforcement Learning against Poisoning Attacks. ICLR 2022.

---

### Official Review · Reviewer_dDco · 2023-10-29

**Soundness:** 3 good
**Presentation:** 4 excellent
**Contribution:** 3 good
**Rating:** 6
**Confidence:** 4

**Summary:**

This paper proposes Robust IQL (RIQL), an offline RL algorithm that works decently even when the dataset is corrupted. It first experimentally and theoretically shows that IQL is robust to dataset noise. Adding upon the discovery, it presents three heuristics that can further improve IQL's performance on the corrupted dataset: (1) observation normalization, (2) Huber loss for value function learning, (3) using the $\alpha$-quantile of an ensemble of Q functions as the target value. The authors conducted experiments on varying the degree and type of corruption and showed that RIQL outperforms other baselines in most of the settings.

**Strengths:**

The paper tackles a novel problem where the offline dataset is corrupted. It provides an exact error bound of IQL under the data contamination setting supported by a mathematically sound argument. The paper also presents an interesting discovery that the Q target distribution has heavy tails and suggests a simple but clever solution, which is to use the Huber loss instead of an MSE loss. To prove the effectiveness of RIQL, multiple experiments were conducted, together with a thorough ablation study. Finally, the paper is overall well-written and easy to understand.

**Weaknesses:**

1. The paper provides no plausible scenarios where a malicious attack on the offline dataset would occur.

2. The definition of $\pi_{\mathcal{D}}(a\mid s)$ is unclear.

3. The paper assumes that IQL can learn the optimal value function $V^*$ from the corrupted dataset without justification.

4. The random corruption setting used in the experiments is a bit unrealistic.

    * If environmental noise exists, the entire dataset would be corrupted, not just a tiny portion as assumed in the experiments.

    * Most offline RL datasets are collections of trajectories not $(s, a, r, s')$ 4-tuples. Adding random noise just to $s$ or $s'$ does not seem to make much sense.

5. The adversarial corruption setting is not really "adversarial" towards algorithms other than EDAC since the adversarial noise was computed via projected gradient descent with respect to the Q functions learned by EDAC. To see whether RIQL is robust to malicious attacks, the adversarial noise should be computed with respect to $Q_\alpha$ learned by RIQL.

6. The paper does not contain experiments for sequence-modeling-based algorithms such as Decision Transformer (Chen et al., 2021) or Diffuser (Janner et al., 2022). As the sequence modeling approach is one of the main branches of offline RL, I believe they should be included.

### Minor comments

1. §4.3 first paragraph "This is evidenced in Figure 5(b), where the **penalty** for attacked data is larger..." ⇒ The term "Penalty" is used before being defined.

2. §C.1 Eq. (11)  third equality ⇒ swap $\tilde{\pi}_E$ and $\pi_E$

3. §C.1 "and the last inequality is obtained by **Cauchy–Schwarz inequality**" ⇒ Cauchy–Schwarz inequality → Hölder's inequality

4.  §C.1 Eq. (15)  ⇒ $\pi_\mu(a\mid s_i)$ → $\pi_{\mathcal{D}}(a\mid s_i)$

5. §C.1 Eq. (16)   ⇒ $\zeta$ → $\zeta_i$

### References

1. Chen, Lili, et al. "Decision transformer: Reinforcement learning via sequence modeling." Advances in neural information processing systems 34 (2021): 15084-15097.

2. Janner, Michael, et al. "Planning with diffusion for flexible behavior synthesis." arXiv preprint arXiv:2205.09991 (2022).

**Questions:**

1. The main theorem holds for any algorithm that uses AWR to learn the optimal policy. Is it possible to add experimental results of other algorithms based on AWR?

2. Does observation normalization also improve the performance of other algorithms?

3. The sentence "This is likely because normalization can help to ensure that the algorithm's performance is not unduly influenced by larger-scale features." from §4.1 is difficult to understand.

---

> ### Author Response · Authors · 2023-11-17
> **Response to Reviewer dDco (part 1/3)**
>
> We thank the reviewer for the constructive comments, and we provide clarification to your concerns as follows. We would appreciate it if you have any further questions or comments.
>
> **Q1: The paper provides no plausible scenarios where a malicious attack on the offline dataset would occur.**
>
> **A1:** To investigate a more challenging setting than the random corruption, we introduce the adversarial corruption, which modifies the values of state, action, reward, and next-state in an adversarial manner. This setting is particularly relevant in scenarios where malicious attackers exist. For example, some annotators might intentionally provide wrong responses for RLHF data collection or assign higher rewards for harmful responses, potentially rendering the LLM harmful for human use when learning from these annotated data. We include the example in Section 1.
>
> **Q2: The definition of $\pi_{D}(a|s)$ is unclear.**
>
> **A2:** Thank you for your question. The policy distribution of the corrupted offline dataset is denoted by $\pi_{\mathcal{D}}$, and its formal definition in a discrete form is given by:
>
> $$
> \pi_{\mathcal{D}}(a|s) = \frac{\sum_{i = 1}^N\mathbf{1}\\{s_i = s, a_i = a\\}}{\max\\{1, \sum_{i = 1}^N\mathbf{1}\\{s_i = s\\}\\}}.
> $$
>
> Here, $\mathbf{1}$ is an indicator function, and $N$ is the number of samples in dataset $\mathcal{D}$. Correspondingly, $\pi_{\mu}(a|s)$ is the behavior policy that induces the clean data. We have clarified this in the revision.
>
> **Q3: The paper assumes that IQL can learn the optimal value function $V^{*}$ from the corrupted dataset without justification.**
>
> **A3:** As IQL's policy and value training can be separated, the analysis of obtaining the optimal value function $V^*$ and weighted imitation policy learning can be considered as two disjoint parts, with the latter being our primary focus. In fact, we have also demonstrated that Huber regression can effectively find the optimal value function even in the presence of the heavy-tailed issue (the original IQL paper proves this in the uncorrupted setting without the heavy-tailed issue). Please refer to the discussion in Appendix C.2 for details. We have revised the main paper to provide justification for this.
>
>
> **Q4: (1) If environmental noise exists, the entire dataset would be corrupted, not just a tiny portion. (2) Adding random noise just to $s$ or $s'$ does not seem to make much sense.**
>
>  **A4:** Thank you for the valuable questions. Regarding the first question, our rationale is based on the fact that in many real-world scenarios, engineers often encounter a combination of high-quality data and poor-quality data. **This mixed data setting is common in robust imitation learning [1][2] and aligns with the $\epsilon$-Contamination model in theoretical analysis [3]. In Section 6.2, we have investigated varying corruption rates from 0 to 50$\\%$, which is not a tiny portion.**  In cases where the entire dataset can be corrupted, it can be considered a special case in our setting by adjusting the corruption rate to 1.0.
>
> Regarding the second question, we remark that the use of full trajectories as the offline dataset is the typical choice in the context of finite-horizon MDPs (e.g., [3]). However, under the framework of discounted MDPs, storing offline RL datasets as independent 4-tuples is a commonly used approach in both theoretical literature [4][5][6][7] and empirical works [8][9]. **Particularly in TD learning, states $s$ and next-states $s'$ play distinct roles: the former defines the input distribution, while the latter determines the future outcome. Consequently, they can be independently corrupted.** In our experiments, we also observed different effects when corrupting $s$ and $s'$: the Huber loss can alleviate dynamics corruption, but it cannot mitigate observation corruption.

---

> ### Author Response · Authors · 2023-11-17
> **Response to Reviewer dDco (part 2/3)**
>
> **Q5: The adversarial corruption setting is not really "adversarial" towards algorithms other than EDAC. ... the adversarial noise should be computed with respect to $Q_{\alpha}$ learned by RIQL.**
>
> **A5:** The adversarial corruption is not specific for EDAC, as we utilize pretrained Q functions on the clean dataset rather than the Q functions obtained during EDAC training on the corrupted dataset. Our motivation for the adversarial corruption is to perform projected gradient descent on an oracle Q function, creating a more challenging setting than random corruption. **In our preliminary experiments shown below, we assessed IQL under adversarial attacks using Q functions trained by CQL, MSG, EDAC, and IQL itself. We observed that EDAC's Q functions yield the most effective attack, while IQL's exhibit the weakest attack performance.** We speculate that this is due to value-based methods learning better value functions for attacking purposes. Consequently, we opted to use pretrained EDAC's Q functions to perform the attack.
>
> | Algorithm for corruption |      IQL      |      CQL      |      MSG      |      EDAC      |
> |:------------------------:|:------------:|:------------:|:------------:|:-------------:|
> | IQL Performance ↓      | 2964.3±477.9 | 1153.8±333.7 | 1528.8±278.4 | **763.9±203.7** |
>
>
> **Q6: The paper does not contain experiments for sequence-modeling-based algorithms such as Decision Transformer (Chen et al., 2021) or Diffuser (Janner et al., 2022).**
>
> **A6:** Thank your for the suggestion. To solve the reviewer's concern, we (1) add an additional discussion in Appendix A regarding algorithms that utilize transformers and diffusion models, and (2) include the results of Decision Transformer under both random and adversarial corruption (Tables 9 and Table 10 in Appendix G). Despite DT leveraging trajectory history information and taking 3.1 times the epoch time compared to RIQL, it still underperforms RIQL by a large margin. For a quick overview, we also provide the average results across three environments (Halfcheetah, Walker2d, and Hopper) below.
>
> | Attack Method | Attack Element |       DT       |     RIQL     |
> |:-------------:|:--------------:|:-------------:|:------------:|
> |     Random    |  Observation   | **51.1±6.3** |  39.4±4.0  |
> |     Random    |     Action     |  20.7±1.8  | **72.7±3.2** |
> |     Random    |     Reward     |  60.9±3.7  | **70.5±5.4** |
> |     Random    |    Dynamics    |  51.1±6.3  | **57.6±3.4** |
> |  Adversarial  |  Observation   |  50.2±7.5  | **52.2±5.7** |
> |  Adversarial  |     Action     |  13.4±1.3  | **53.8±4.5** |
> |  Adversarial  |     Reward     |  56.8±6.2  | **65.0±4.0** |
> |  Adversarial  |    Dynamics    |  50.2±7.5  | **54.0±15.0** |
> |    Average    |                |     44.3     |   **58.1**   |
>
>
> **Q7: Minor comments**
>
> **A7:** Thank you for pointing out these typos. We have made revisions in our updated manuscript.
>
> **Q8: The main theorem holds for any algorithm that uses AWR to learn the optimal policy. Is it possible to add experimental results of other algorithms based on AWR?**
>
> **A8:** Yes, we add experimental results of AWAC [10], which utilizes a weighted imitation learning framework, in Figure 1 to provide additional validation for our theory. As shown in Figure 1, AWAC exhibits notably robust performance under observation and action corruption, despite not quite reaching the level of IQL. These results not only validate the effectiveness of weighted imitation learning in enhancing robustness but also indicate that IQL's expectile regression and detached value training contribute to further improvements, which is also evidenced by the ablation study in Appendix E.1.

---

> ### Author Response · Authors · 2023-11-17
> **Response to Reviewer dDco (part 3/3)**
>
> **Q9: Does observation normalization also improve the performance of other algorithms?**
>
> **A9:**
> Thank you for your question. We implemented observation normalization for both EDAC and CQL on the Walker2d and Hopper tasks, incorporating random data corruption across four elements. The average results, as displayed below, indicate a modest overall improvement for these algorithms.
>
>
> | Environment | Attack Element | EDAC         | EDAC(norm)   | CQL          | CQL(norm)    |
> |-------------|----------------|--------------|--------------|--------------|--------------|
> | Walker2d    | observation    | -0.2±0.3     | -0.1±0.1     | 19.4±1.6     | 33.9±8.9     |
> |       Walker2d      | action         | 83.2±1.9     | 90.8±1.8     | 62.7±7.2     | 67.2±8.0     |
> |      Walker2d       | reward         | 4.3±3.6      | 0.7±1.2      | 69.4±7.4     | 65.3±5.7     |
> |     Walker2d        | dynamics       | -0.1±0.0     | -0.2±0.0     | -0.2±0.1     | -0.3±0.1     |
> | Hopper      | observation    | 1.0±0.5      | 3.2±1.7      | 42.8±7.0     | 45.4±9.1     |
> |  Hopper            | action         | 100.8±0.5    | 100.6±0.2    | 69.8±4.5     | 64.0±9.1     |
> |   Hopper           | reward         | 2.6±0.7      | 4.7±4.3      | 70.8±8.9     | 61.8±14.5    |
> |    Hopper          | dynamics       | 0.8±0.0      | 0.8±0.0      | 0.8±0.0      | 0.9±0.1      |
> | Average score |              | 24.1         | 25.1         | 41.9         | 42.3         |
>
>
>
> **Q10: The sentence "This is likely because normalization can help to ensure that the algorithm's performance is not unduly influenced by larger-scale features." is difficult to understand.**
>
> **A10:** Thank you for pointing out this. To make it clear, we polish the sentence to "Normalization is likely to help ensure that the algorithm's performance is not unduly affected by features with large-scale values.".
>
>
> **References**
>
> [1] Sasaki F, Yamashina R. Behavioral cloning from noisy demonstrations. ICLR 2020.
>
> [2] Liu L, et al. Robust imitation learning from corrupted demonstrations[J]. ArXiv 2022.
>
> [3] Jin Y, et al. Is pessimism provably efficient for offline rl? ICML 2021.
>
> [4] Zhang X, et al. Corruption-robust offline reinforcement learning. AISTATS 2022.
>
> [5] Rashidinejad P. et al. Bridging Offline Reinforcement Learning and Imitation Learning:
> A Tale of Pessimism.
>
> [6] Xie T. et al. Bellman-consistent Pessimism for
> Offline Reinforcement Learning
>
> [7] Uehara M and Sun W. Pessimistic Model-based Offline Reinforcement Learning under Partial
> Coverage.
>
> [8] Kumar A, et al. Conservative q-learning for offline reinforcement learning. NeurIPS 2020.
>
> [9] An G, et al. Uncertainty-based offline reinforcement learning with diversified q-ensemble. NeurIPS 2021.
>
> [10] Nair A, et al. Awac: Accelerating online reinforcement learning with offline datasets. arXiv 2020.

---

> > ### Comment · Reviewer_dDco · 2023-11-20
> >
> > Thank you for the detailed response. Here are some questions I have regarding your response.
> >
> > A2: Under this definition, $\pi_{\mathcal{D}}$ becomes a discrete distribution. How would you define the ratio between a discrete distribution $\pi_{\mathcal{D}}$ and a continuous distribution $\pi_{\mu}$?
> >
> > A4-2: I agree that regarding the dataset as a collection of 4-tuples can be helpful for the learning process, but in terms of actually collecting and storing the data, I feel the trajectory approach is just superior. First, the environmental interactions are collected in terms of trajectories, not $(s, a, r, s')$ 4-tuples. Second, trajectory-based storage is more memory efficient. Lastly, it is easy to convert a collection of trajectories into a collection of 4-tuples, but not the other way around. I can't think of any reason why one would adopt the 4-tuple approach to create an offline RL DB in practice.
> >
> > A10: What do you exactly mean by "larger-scale features"? Why does normalization help reduce their influence?

---

> > > ### Author Response · Authors · 2023-11-21
> > > **Thanks for your response**
> > >
> > > Thank you for your prompt reply. We are pleased to address any remaining concerns you may have.
> > >
> > > **Q2: Under this definition,  becomes a discrete distribution. How would you define the ratio between a discrete distribution  and a continuous distribution?**
> > >
> > > **A2:** Thank you for your question. Typically, the action space is assumed to be finite for ease of theoretical analysis. In this case, both $\pi_{\mathcal{D}}$ and $\pi_\mu$ are discrete, and hence the ratio is well-defined. We have clarified this in the revision. Besides, we can also assume that we have a clean dataset $\hat{D}=\\{ (\hat{s}\_i, \hat{a}\_i,\hat{r}\_i,\hat{s}'\_i)     \\}\_{i=1}^N$ and define the corresponding conditional distributions $\pi_{\hat{\mathcal{D}}}$. In this scenario, the ratio is also well-defined.
> > >
> > >
> > > **Q4-2: I agree that regarding the dataset as a collection of 4-tuples can be helpful for the learning process, but in terms of actually collecting and storing the data, I feel the trajectory approach is just superior ... I can't think of any reason why one would adopt the 4-tuple approach to create an offline RL DB in practice.**
> > >
> > >
> > > **A4-2:** As the reviewer agreed, the 4-tuple approach is advantageous for the learning process, and many current offline RL applications are learning-centric. While data can be collected in trajectory form, it is often transformed into 4-tuples prior to training. The 4-tuple approach has its own advantages, particularly in learning-centric settings where data collection and storage are not the primary bottlenecks. These merits include:
> > >
> > > * 1. Overall Memory efficiency: In the 4-tuple approach, each experience is stored independently, which can be **more memory-efficient when dealing with trajectories of varying lengths**. This is because you don't need to allocate memory based on the length of the longest trajectory, which can be wasteful if the length of trajectories significantly vary.
> > >
> > > * 2. Processing Efficiency: In the context of batch learning algorithms, which are commonly used in offline RL, **constructing fixed-size batches is straightforward with random sampling from a collection of 4-tuples**. Conversely, a trajectory-based approach can introduce additional computational overhead when constructing batches, especially when trajectories have different lengths.
> > >
> > > * 3. Flexibility: The 4-tuple method can be **easily extended to sample-level importance sampling**, emphasizing a few critical actions in long trajectories. Moreover, **it allows for simple removal of unsafe or unethical actions**, such as harmful responses in LLMs, while preserving other useful interactions within a trajectory.
> > >
> > >
> > >
> > > While the trajectory-based approach has its advantages, the 4-tuple approach can also be beneficial in certain contexts. The choice between the two should be guided by the specific requirements of your task, the characteristics of your data, and the RL algorithm you're employing. Considering that a large number of offline RL algorithms (including IQL, CQL, EDAC, SQL, and MSG) follow the 4-tuple approach, we have chosen to focus on studying data corruption in this setting.
> > >
> > >
> > > **Q10: What do you exactly mean by "larger-scale features"? Why does normalization help reduce their influence?**
> > >
> > > **A10:** By "larger-scale features", we refer to the dimensions within the state space that have a broader range of values. During training, the computation of gradients is dependent on the input values, causing gradients for larger-scale features to be larger than those for smaller-scale features. Corruption noise, magnified by each dimension's standard deviation, can worsen this issue, further emphasizing larger-scale dimensions in learning.
> > >
> > > Normalization mitigates these issues by rescaling features to a standard range, ensuring equal contribution to learning. This reduces the influence of larger-scale dimensions, promoting a balanced and robust learning process.
> > >
> > > Thanks again for your time and effort!

---

> ### Comment · Reviewer_dDco · 2023-11-22
>
> Thank you for your reply. I want to make some clarifications regarding **Q4-2**, though.
>
> 1. We can preserve the sequential information without allocating memory based on the longest trajectory by saving the data as a sequence of (observation, action, reward, truncated, terminated) 5-tuple. This is exactly how D4RL stores data.
>
> 2. For MDPs, the 4-tuple view is sufficient, but for POMDPs, we need to preserve the sequential information. Most real-world problems are not Markovian, so the sequential information has to be preserved.
>
> To sum up, even if an algorithm adopts the 4-tuple approach, it is better to store the data as a sequence of (observation, action, reward, truncated, terminated) 5-tuples and then convert it to a collection of $(s, a, r, s')$ 4-tuples in memory during the learning process. As attacks on the memory values are extremely difficult, I think assuming corruption would happen independently on s and s' is unrealistic.

---

> > ### Author Response · Authors · 2023-11-23
> > **Response to Reviewer dDco**
> >
> > Thank you for providing a detailed explanation of your remaining concerns. Your question prompts us to consider the realistic scenarios in which data corruption can occur.
> >
> > We agree with you on the benefits of saving data into 5-tuples and then converting it to a collection of 4-tuples. However, we would like to clarify a few points:
> >
> > * 1. In our paper, **we focus on MDP rather than POMDP, making the 4-tuple approach a natural choice**, especially when storage is not the primary bottleneck. Exploring data corruption in POMDP scenarios could be an interesting topic for future research.
> >
> > * 2. Importantly, **data corruption can occur at any stage** beyond just storage, such as writing, reading, transmission, or processing of data. While memory corruption has a lower probability of corruption compared to disk storage, we cannot guarantee 0\% error rate. There have been studies on data corruption in memory [1][2][3][4], such as silent data corruption, programming errors, and even intentional modifications by hackers.
> >
> > * 3. Our work aims to address a comprehensive range of data corruption types. Consequently, **conducting an independent analysis of each element's vulnerability effectively highlights the disparities in their susceptibility**.
> >
> > * 4. Our corruption method aligns with the concept of $\epsilon$-contamination in offline RL theoretical analysis [5], which considers arbitrary corruption on the 4-tuples $(s,a,r,s')$. This work also motivates us to adopt this corruption method.
> >
> > **As our work is one of the early studies on offline RL under various types of data corruption, there is no established standard for us to follow**. We genuinely appreciate the reviewer's perspective regarding the dependency between corruption on states and next-states. Furthermore, we provided clarification on the motivations behind our corruption method. **To address the reviewer's remaining concern, we also included a section in the Appendix J to discuss the corruption setting.**
> >
> > Thank you once again for your time and effort in improving our work.
> >
> > **References:**
> >
> > [1] Zhang Z, Huang L, Huang R, et al. Quantifying the impact of memory errors in deep learning[C]//2019 IEEE International Conference on Cluster Computing (CLUSTER). IEEE, 2019: 1-12.
> >
> > [2] Llorente-Vazquez O, Santos I, Pastor-Lopez I, et al. The neverending story: Memory corruption 30 years later[C]. CISIS 2021 and ICEUTE 2021. Springer International Publishing, 2022: 136-145.
> >
> >
> > [3] Meza J, Wu Q, Kumar S, et al. Revisiting memory errors in large-scale production data centers: Analysis and modeling of new trends from the field[C]//2015 45th Annual IEEE/IFIP International Conference on Dependable Systems and Networks. IEEE, 2015: 415-426.
> >
> > [4] Alshmrany K, Bhayat A, Brauße F, et al. Position Paper: Towards a Hybrid Approach to Protect Against Memory Safety Vulnerabilities[C]//2022 IEEE Secure Development Conference (SecDev). IEEE, 2022: 52-58.
> >
> >
> > [5] Zhang X, Chen Y, Zhu X, et al. Corruption-robust offline reinforcement learning[C]//International Conference on Artificial Intelligence and Statistics. PMLR, 2022: 5757-5773.

---

### Official Review · Reviewer_41NY · 2023-10-29

**Soundness:** 3 good
**Presentation:** 3 good
**Contribution:** 2 fair
**Rating:** 6
**Confidence:** 4

**Summary:**

This work concentrates on dealing with diverse types of data corruption associated with state, action, reward, and transition kernel in offline RL history datasets. In particular, it first did data corruption test on some existing offline RL algorithms to show their vulnerable behaviors against data corruption. A theoretical result for the robustness of IQL has been shown w.r.t. the data corruption ratio. Then a robust variant of IQL has been proposed and outperform offline RL baselines, which consist of three key parts: state normalization, Huber loss, and $\alpha$-quantile Q ensemble.

**Strengths:**

1. It shows interesting testing of the effect of data corruption for current offline RL algorithms.
2. Some theoretical relationship between the data corruption ratio and the performance of IQL is presented.
3. A new robust IQL algorithm has been proposed and outperforms conducted baselines when data corruption appears.

**Weaknesses:**

1. Except for the traditional offline RL algorithms based on TD learning or the Bellman operator, there exists some other new baselines using a transformer or a diffusion model. It is helpful to involve the discussion about such algorithms at least.
2. Since the new algorithm (RIQL) involves in ensemble, which is a very powerful trick in RL algorithms, it is better to add some baselines that also use ensemble Q, while not appearing in the experiments if I didn't miss something.
3. There does not exist comparisons between the proposed algorithm RIQL to other existing robust RL algorithms, but only to non-robust counterparts.

**Questions:**

1. Is there any study on ablation study for the proposed RIQL that is similar to Figure 1. It will be helpful to see how can RIQL handle different types of data corruption.

---

> ### Author Response · Authors · 2023-11-17
> **Response to Reviewer 41NY**
>
> We thank the reviewer for the constructive comments, and we provide clarification to your concerns below. We would appreciate it if you have any further questions or comments.
>
> **Q1: other new baselines using a transformer or a diffusion model. It is helpful to involve the discussion about such algorithms at least.**
>
> **A1:** Thank you for the suggestion. We include a discussion in Appendix A regarding algorithms that utilize transformers and diffusion models. Additionally, we compare our method to Decision Transformer (DT) in both the random and adversarial corruption benchmarks, as shown in Tables 9 and Table 10 (Appendix G). Despite DT leveraging trajectory history information and taking 3.1 times the epoch time compared to RIQL, it still underperforms RIQL by a large margin. For a quick overview, we also provide the average scores over three environments (Halfcheetah, Walker2d, and Hopper) below.
>
> | Attack Method | Attack Element |   DT   |  RIQL  |
> |:-------------:|:--------------:|:-----:|:-----:|
> |     Random    |  Observation   | **51.1** |  39.4  |
> |     Random    |     Action     |  20.7  | **72.7** |
> |     Random    |     Reward     |  60.9  | **70.5** |
> |     Random    |    Dynamics    |  51.1  | **57.6** |
> |  Adversarial  |  Observation   |  50.2  | **52.2** |
> |  Adversarial  |     Action     |  13.4  | **53.8** |
> |  Adversarial  |     Reward     |  56.8  | **65.0** |
> |  Adversarial  |    Dynamics    |  50.2  | **54.0** |
> |    Average    |                |  44.3  | **58.1** |
>
> **Q2: add some baselines that also use ensemble Q, while not appearing in the experiments if I didn't miss something.**
>
> **A2:** In our initial submission, we have included SOTA ensemble-based baselines, such as EDAC [1] and MSG [2]. While these baselines demonstrate exceptional performance on clean data, we note that they are considerably more vulnerable to data corruption, with a performance drop of approximately $60\% \sim 70\%$, as presented in Table 1 and Table 2 (Section 6.1).
>
> **Q3: no comparisons between the proposed algorithm RIQL to other existing robust RL algorithms**
>
> **A3:** To the best of our knowledge, prior to our submission, there were no empirical robust offline RL baselines designed for the data corruption setting. Previous research primarily focused on theoretical guarantees [3], certification protocols [4], and evaluations [5] for offline RL with data corruption. However, we find a recent study [6] that introduced a solution called UWMSG for addressing reward and dynamics corruption, and we compare it with our proposed method.
>
> The comparative results can be found in Table 9 and Table 10 (Appendix G). Notably, RIQL demonstrates significantly improved performance compared to UWMSG. We believe that our work will serve as a strong baseline for future research in this setting. For a quick overview, we have also provided the average scores across three environments below.
>
> | Attack Method | Attack Element | UWMSG |  RIQL  |
> |:-------------:|:--------------:|:-----:|:-----:|
> |     Random    |  Observation   |  5.0  | **39.4** |
> |     Random    |     Action     | 65.7  | **72.7** |
> |     Random    |     Reward     | 55.6  | **70.5** |
> |     Random    |    Dynamics    |  9.6  | **57.6** |
> |  Adversarial  |  Observation   |  8.3  | **52.2** |
> |  Adversarial  |     Action     | 26.9  | **53.8** |
> |  Adversarial  |     Reward     | 40.1  | **65.0** |
> |  Adversarial  |    Dynamics    |  4.6  | **54.0** |
> |    Average    |                | 27.0  | **58.1** |
>
> **Q4: Is there any study on ablation study for the proposed RIQL that is similar to Figure 1. It will be helpful to see how can RIQL handle different types of data corruption.**
>
> **A4:** Thank you for the suggestion. In Appendix H, we include an additional ablation study under separate observation, action, reward, and dynamics corruption. These results are helpful for understanding how each component of RIQL contributes to performance under various data corruption scenarios. For example, from the results, we conclude that the Huber loss is more beneficial for reward and dynamics corruption, but less effective for observation and action corruption. Overall, all three components play an important role in enhancing the performance of RIQL.
>
>
> **References**
>
> [1] An G, et al. Uncertainty-based offline reinforcement learning with diversified q-ensemble. NeurIPS, 2021.
>
> [2] Ghasemipour K, et al. Why so pessimistic? estimating uncertainties for offline rl through ensembles, and why their independence matters. NeurIPS, 2022.
>
> [3] Zhang X, et al. Corruption-robust offline reinforcement learning. AISTATS, 2022.
>
> [4] Wu F, et al. COPA: Certifying Robust Policies for Offline Reinforcement Learning against Poisoning Attacks. ICLR 2021.
>
> [5] Li A, et al. Survival Instinct in Offline Reinforcement Learning. NeurIPS 2023.
>
> [6] Ye C, et al. Corruption-Robust Offline Reinforcement Learning with General Function Approximation. NeurIPS 2023.

---

> > ### Comment · Reviewer_41NY · 2023-11-22
> > **Response to the author**
> >
> > Thank you for the author's rebuttal. The answers address my concerns and I would love to keep my positive support for this work.

---

### Official Review · Reviewer_t4o4 · 2023-11-01

**Soundness:** 3 good
**Presentation:** 3 good
**Contribution:** 3 good
**Rating:** 8
**Confidence:** 3

**Summary:**

This paper concerns the robustness of different offline reinforcement learning methods to comprehensive data corruption, including states, actions, rewards, and dynamics. Their empirical results show that IQL method has better robustness to all types of data corruptions except noisy dynamic. The authors first give a theoretical explanation about IQL's robust performance, and according to empirical evidence, attribute this exceptional phenomenon to the heavy-tailed Q-targets. To verify this assumption and address this issue, this paper adopts observation normalization and Huber loss function for robust value function learning. Besides, this paper finds the potential negative value exploding issue of the clipped double Q-learning technique used in the original IQL and adopts quantile Q estimators instead of the LCB estimation. All the above modifications are verified to improve the performence of IQL under the data corruption of enviorment dynamic.

**Strengths:**

1. This paper provides comprehensive analysis and interesting findings on different types of data corruption in the offline RL;
2. Sufficient empirical evidence on the possible reason of the failure setting, and efficient solutions to address these issues.

**Weaknesses:**

1. Lack theoretical evidence on heavy-tailed Q-target issue;
2. Heavy work on parameter finetuning;

**Questions:**

In Fig.1 only two scales are illustrated, would you please make it more clear about the underlying mechanism that generates the corrupted data in each cases,  and how to ensure that the generated noisy data is realistic and representative in the sense that they do represent typical noisy data we encountered in practice? Intuitively, different level of noise would lead to different robustness outcomes of various algorithms - in this sense, how do you ensure that the empirical evidence is conclusive?

---

> ### Author Response · Authors · 2023-11-17
> **Response to Reviewer t4o4**
>
> We thank the reviewer for the constructive comments, and we provide clarification to your concerns below. We would appreciate it if you have any further questions or comments.
>
> **Q1: Lack the theoretical evidence on heavy-tailed Q-target issue**
>
> **A1:** The identification of the heavy-tailed issue primarily stems from empirical observations, including visualization and the calculation of Kurtosis values. It is challenging to provide a rigorous mathematical explanation for the cause of the heavy-tailed issue. However, we offer some intuitive speculation for this issue — dynamics corruption can corrupt $s'$ to less frequently visited or even artificial states $\hat s'$, whose values $V_{\psi}(\hat s')$ exhibit higher uncertainty, thereby increasing the variance of the Q-target during offline training. Furthermore, we want to underscore that many existing works [1][2] in this field also identify heavy-tailed problems empirically and adopt assumptions similar to Assumption 4 in our paper. This assumption is mild in that it only requires the existence of $(1+\nu)$-th moment, and our theoretical analysis (Lemma 5, Section 5.2)  demonstrates why Huber regression can effectively address the heavy-tailed issue.
>
> **Q2: Heavy work on parameter finetuning**
>
> **A2:** Due to varying inherent properties of different attacks and environments, optimal hyperparameters differ. However, after conducting a parameter search, we find that in most settings, $N=5, \alpha\in\{0.1, 0.25\}, \delta\in\{0.1,1.0\}$ yields the best or comparable performance. This hyperparameter space is relatively small. Additionally, we provide a hyperparameter study in Appendix E.8 to guide hyperparameter tuning. For instance, it is recommended to use larger $\delta$ (e.g., $1.0$) to address dynamics and reward corruption, as opposed to smaller values (e.g., $0.1$) for observation and action corruption.
>
> **Q3: making it more clear about the mechanism that generates the corrupted data in each case, and how to ensure that the generated noisy data represent typical noisy data in practice?**
>
> **A3:** In Appendix D.1, we present a comprehensive explanation of the generation of corrupted data, where we consider two major forms of corruption: random corruption and adversarial corruption. Random corruption is similar to random noise encountered in measurement due to the inherent variance of the device or external factors in the environment. To simulate this scenario, we introduce Gaussian noise for states, actions, and next-states. To investigate a more challenging setting, we introduce adversarial corruption, which modifies the values of state, action, reward, and next-state in an adversarial manner. This setting is particularly relevant in scenarios where malicious attackers exist. For example, some annotators might intentionally provide wrong responses for RLHF or assign higher rewards for harmful responses, potentially making the LLM harmful for human use when learning from the annotated data. The two types of corruption also align with prior related works in robust imitation learning [3] and robust RL [4].
>
> **Q4: different levels of noise would lead to different robustness outcomes - how do you ensure that the empirical evidence is conclusive?**
>
> **A4:** We agree that different levels of noise can yield different robustness outcomes. In our setting, the noise level is determined by two factors: the corruption scale (the magnitude of each added noise) and the corruption rate (the proportion of data that is corrupted). **In Section 6.2, we assess the performance of different algorithms across varying corruption rates (0$\%$ $\sim$ 50$\%$)** and demonstrate that RIQL exhibits greater robustness compared to other algorithms at different corruption rates. Additionally, we present results for random and adversarial corruption at scale 1 in Section 6.1 and scale 2 in Appendix E.6. Our findings indicate that RIQL consistently outperforms other algorithms. Considering that many baselines already attain scores lower than 10 under corruption scale 2, which is a notably low score, and taking into account the fact that noise with too large scale can be more easily detected in real-world scenarios, we have not continued to increase the corruption scale.
>
> **References**
>
> [1] Garg S, et al. On proximal policy optimization’s heavy-tailed gradients. ICML, 2021.
>
> [2] Zhuang V, Sui Y. No-regret reinforcement learning with heavy-tailed rewards. AISTATS, 2021.
>
> [3] Sasaki F, Yamashina R. Behavioral cloning from noisy demonstrations. ICLR, 2020.
>
> [4] Zhang H, et al. Robust deep reinforcement learning against adversarial perturbations on state observations. NeurIPS, 2020.

---

> > ### Comment · Reviewer_t4o4 · 2023-11-23
> > **Thank you for your response.**
> >
> > My concerns are well addressed and I will maintain my previous score.

---

### Author Response · Authors · 2023-11-17
**Rebuttal Summary**

We would like to thank the reviewers for their valuable time and effort invested in reviewing our work. We genuinely appreciate the constructive feedback, and we are encouraged by the recognition of the novelty and contribution of our work. We have provided detailed responses and clarifications for each reviewer's comments.

During the rebuttal phase, we revised our manuscript to address reviewers' concerns. These modifications have been highlighted in blue. A summary of the revisions is listed below:

1. **[More baselines]** We added a sequence modeling baseline, DT, and a recent robust offline RL baseline, UWMSG, in Appendix G.

2. **[Additional Ablation]** We supplemented additional ablation study of RIQL under separate corruption in Appendix H.

3. **[AntMaze Tasks]** We included additional evaluation results on AntMaze tasks under data corruption in Appendix I.

4. **[Manuscript Revision]** We revised the manuscript according to reviewers' feedback, such as including a discussion of sequence-modeling baselines in Appendix A, adding a malicious corruption example in Section 1, and fixing typos.

---

### Meta-Review · Area_Chair_jvpo · 2023-12-11

**Metareview:**

This paper proposes a comprehensive investigation into the performance of offline RL algorithms under various data corruptions, highlighting the resilience of implicit Q-learning (IQL) compared to other methods. Through empirical and theoretical analyses, the authors identify the supervised policy learning scheme as key to IQL's robustness. To further enhance its performance, they introduce Robust IQL (RIQL), incorporating Huber loss and quantile estimators to handle heavy-tailed targets and balance penalization for corrupted data, resulting in a more robust offline RL approach.
Generally, this paper studies an important problem of offline RL, i.e., improving the robustness of offline RL. The authors provide comprehensive analysis and innovative findings on different types of data corruption in the offline RL, making this paper interesting and of high tech contribution.
The discussions are in-depth and all the reviewers reach an agreement to accept this paper.

**Justification For Why Not Higher Score:**

The paper may still be imperfect in several aspects such as the heavy hyperparameter tuning and the lack of justification of the practical scenarios of malicious attacks in offline RL settings.

**Justification For Why Not Lower Score:**

This paper studies an important problem of offline RL, i.e., improving the robustness of offline RL. The authors provide comprehensive analysis and innovative findings on different types of data corruption in the offline RL, making this paper interesting and of high-tech contribution, which deserves a talk in ICLR 2024.

---

### Decision · Program_Chairs · 2024-01-16

Accept (spotlight)